# Flowers: A Warp Drive for Neural PDE Solvers

**Till Muser** [1]   **Alexandra Spitzer** [1]   **Matti Lassas** [2]   **Maarten V. de Hoop** [3]   **Ivan Dokmanić** [1]

## Abstract

We introduce FLOWERS, a neural architecture for learning PDE solution operators built entirely from multihead warps. Aside from pointwise channel mixing and a multiscale scaffold, FLOWERS use no Fourier multipliers, no dot-product attention, and no convolutional mixing. Each head predicts a displacement field and warps the mixed input features. Motivated by physics and computational efficiency, displacements are predicted pointwise, without any spatial aggregation, and nonlocality enters *only* through sparse sampling at source coordinates, *one* per head. Stacking warps in multiscale residual blocks yields FLOWERS, which implement adaptive, global interactions at linear cost. We theoretically motivate this design through three complementary lenses: flow maps for conservation laws, waves in inhomogeneous media, and a kinetic-theoretic continuum limit. FLOWERS achieve excellent performance on a broad suite of 2D and 3D time-dependent PDE benchmarks, particularly flows and waves. A compact 17M-parameter model consistently outperforms Fourier, convolution, and attention-based baselines of similar size, while a 150M-parameter variant improves over recent transformer-based foundation models with much more parameters, data, and training compute.

## 1. Introduction

Deep neural networks have become credible surrogate solvers for PDEs. After training, they run fast, they are differentiable, and they can be conditioned on parameters, forcing, or geometry. What is less settled is what it takes for such models to behave physically rather than as data interpolants, to capture qualitative structure, roll out stably, and remain effective across physical regimes.

Most neural PDE backbones rely on Fourier mixing, convolutions, or attention. Fourier layers and attention provide global coupling, but treat long-range interactions as equally plausible a priori. Convolutions build a locality prior, but interactions are simple linear stencils with weak expressivity at the fringe of the receptive field. For many conservative phenomena, the evolution of the conserved quantities $u(t, x)$ at a point depends primarily on their pointwise coupling at $x$ and on a small number of "upstream" contributions at earlier times, sampled along characteristics or rays, in an adaptive, state-dependent way.

Here we introduce FLOWERS, a neural PDE architecture built (near-)exclusively from learned non-linear coordinate warps (or pullbacks). At each target location $x$, we predict a set of per-head displacements $(\varrho^{(h)}(x))_{h=1}^{H}$ using only pointwise information at $x$, and induce nonlocal interaction by sampling features at the displaced coordinates $x + \varrho^{(h)}(x)$. Heads are mixed pointwise and composed in depth and scale. With a fixed number of heads the cost is linear in the number of grid points, which makes it easy to scale to 3D. We find, perhaps surprisingly, that coordinate warps are a sufficient and scalable primitive to build state-of-the-art neural PDE solvers.

One way to see why warps are a good mechanism for learning physics is via the wave equation. In the high-frequency regime, the solution operator is a Fourier integral operator (FIO): it routes localized wave packets along rays (see Figure 2 and Appendix E), which can be read as a sum of pullbacks, or warps, of the initial data. This structure motivated earlier work on neural networks for waves (Kothari et al., 2020) which showed promising results, but only in a narrow, stylized class of wave-related problems with limited practical reach. This pattern is common in scientific ML: closely mirroring the mathematical structure of an operator class does not necessarily result in strong inductive biases for learning general PDE solution maps. Here our goal is to extract the key computational primitives from this line of work, guided by physics, and to complement it with careful deep-learning engineering.

---

[1]Department of Mathematics and Computer Science, University of Basel, Basel, Switzerland [2]Department of Mathematics and Statistics, University of Helsinki, Helsinki, Finland [3]Simons Chair in Computational and Applied Mathematics and Earth Science, Rice University, Houston, TX, USA. Correspondence to: Till Muser <till.muser@unibas.ch>, Ivan Dokmanić <ivan.dokmanic@unibas.ch>.

*Proceedings of the 43rd International Conference on Machine Learning*, Seoul, South Korea. PMLR 306, 2026. Copyright 2026 by the author(s).

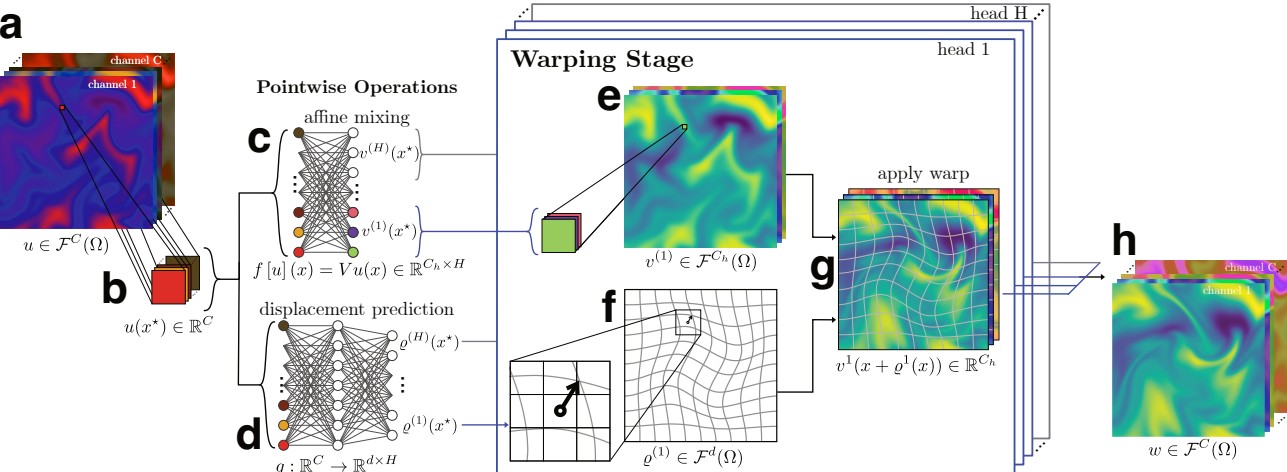

*Figure 1.* The multi-head SELFWARP layer. (a) Given an input field $u$ with $C$ channels, (b) at each location $x$ the layer computes (c) a local per-head value $v^{(h)}(x) \in \mathbb{R}^{C_h}$ via a linear (affine) projection of the input $u(x)$ and (d) a displacement field (one per head) $\varrho^{(h)}(x)$, also pointwise, using a small MLP. Each $C_h$-channel per-head value field $v^{(h)}$ (e) is warped along the same per-head displacement field $\varrho^{(h)}$ (f) to compute the warped head output (g). The outputs of all heads are concatenated at the output (h).

Warps are not new in deep learning, but they appear as augmentations of more complex pipelines which involve convolution and/or attention layers (similarly as with FIO networks). A canonical example is the spatial transformer (Jaderberg et al., 2015). The same sampling primitive underlies deformable convolutions (Dai et al., 2017), and reappears in deformable transformers (Zhu et al., 2021), where each query predicts a set of sampling offsets, which is shown to improve encoder–decoder object detection transformers. Related ideas have been used for dynamic upsampling (Liu et al., 2023), image registration (Balakrishnan et al., 2019), and scientific forecasting (Zakariaei et al., 2024); in all cases, warping is a component of a convolutional or transformer pipeline.

Motivated by physics and compute efficiency, FLOWERS adopt a different perspective. We make warping the main and *only* adaptive nonlocal mixing primitive: each layer predicts a *single* per-head displacement $\varrho^{(h)}(x)$ using only *pointwise* information at the target location $u(x)$, and induces nonlocal interaction by sampling features at $x + \varrho^{(h)}(x)$, one displacement per head. This keeps the operation lightweight, scalable, and grounded in characteristic structure of PDEs, with nonlocality entering only through sparse pullbacks. We then build a modern backbone around this primitive—multi-head organization, residual connections, and a U-Net-like multiscale scaffold[1]. Empirically, this single primitive is enough to obtain state-of-the-art neural PDE solvers across 2D and 3D benchmarks, without Fourier multipliers or dense dot-product attention.

As we show, even a modestly sized FLOWER outperforms

strong Fourier-, convolutional-, and attention-based baselines across diverse benchmarks spanning fluid dynamics, wave propagation, and complex geometries. On the *The Well* benchmark suite (Ohana et al., 2024), 17M-parameter FLOWERS achieve state-of-the-art supervised performance while being efficient to train and run. Scaling to ∼100–150M parameters closes the gap to, and even improves over, very recent foundation-scale PDE models trained with far more parameters, data, and compute. (On several datasets even our "tiny" model achieves stronger performance.) This makes FLOWERS a plausible new family of scientific ML backbones, with considerable headroom: we believe that performance will only improve as the community explores the new design space.

### 1.1. Contributions

(i) We introduce the SELFWARP layer and the FLOWER architecture, showing how to build effective neural PDE solvers from lightweight pointwise warps.

(ii) We connect the design of FLOWERS to characteristic/ray structure of solution operators to key PDEs and to a natural kinetic theory (phase space) limit.

(iii) We show that with a U-Net-like scaffold, FLOWERS outperform strong baselines across datasets and regimes including on almost entire "The Well".

(iv) We show that FLOWERS easily scale to 3D. We also show that increasing the number of parameters to 150M makes them competitive with much larger and more resource-intensive foundation models.

Finally, we note that the design of FLOWERS makes them more interpretable than generic architectures. In Figure 6 we show an example of a warp field predicted by a *trained*

---

[1]Code available on GitHub: github.com/t-muser/flowers

FLOWER. Strikingly, the learned displacement field itself organizes into vortices, mirroring the rotational structure of the underlying flow.

## 2. The Flower Architecture

The design of FLOWERS is inspired by the structure of PDEs known as systems of conservation laws. Such systems are very general and include hydrodynamics (e.g. the Navier–Stokes equations). While the resulting architecture applies well beyond conservation laws, they provide a clean narrative to describe the core mechanics.

We write $u(t, x) \in \mathbb{R}^C$ for the vector of $C$ "conserved quantities", also known as the state, whose evolution we want to predict; for compressible Navier–Stokes, $u$ comprises density, momentum, and energy. A prototypical system is[2]

$$\partial_t u(x,t) + \nabla_x \cdot G(u(x,t)) = 0,$$

where $G(u)$ are the so-called fluxes which describe how the conserved quantities flow in space. As we show in Section 3.1, a key component of the solution operators for such systems is a simple pullback, or warp. We now introduce this basic component and the resulting architecture before moving to a more substantial theoretical motivation.

### 2.1. Architecture

For clarity we present the architecture in continuous coordinates. Let $\Omega \subset \mathbb{R}^d$ be the domain of interest and $\mathcal{F}^C(\Omega) := \{u : \Omega \to \mathbb{R}^C\}$ the space of $C$-channel fields on $\Omega$. Given a $\tau : \Omega \to \mathbb{R}^d$ we define the pullback $(\tau^* v)(x) := v(\tau(x))$. For a $C$-channel field $u$ and a finite-dimensional map $h : \mathbb{R}^C \to \mathbb{R}^{C'}$, we define $(h[u])(x) := h(u(x))$; that is to say, $h[\cdot]$ acts on $u$ pointwise. Concatenation along the channel dimension is denoted $(u \oplus v)(x) = u(x) \oplus v(x)$.

**Multihead warp (pullback layer)**  The main component of FLOWERS is a multihead SELFWARP layer. We begin with a single head. The layer SELFWARP $: \mathcal{F}^{C_{\text{in}}}(\Omega) \to \mathcal{F}^{C_{\text{out}}}(\Omega)$ maps a $C_{\text{in}}$-channel input field $u$ to a $C_{\text{out}}$-channel output field $w$ as

$$w(x) = \text{SELFWARP}_{V,g}[u](x)$$
$$\stackrel{\text{def}}{=} V u(x + \varrho(x))$$

where $\varrho(x) = g(u(x))$; in concise pullback notation, SELFWARP maps

$$u \mapsto w = (\text{Id} + \varrho)^*(Vu).$$

Crucially, the map $u \mapsto \varrho$ is local in a strong sense, i.e., pointwise: $\varrho(x)$ depends on $u(x)$ but not on $u(x')$ for $x' \neq$

[2]Perhaps augmented with source and diffusive terms.

$x$. The warping itself is nonlocal and nonlinear, but sparse, as it samples the input at a single point.

A single deformation has limited expressivity so we build a multihead variant in the spirit of multihead attention. Fix $H$ heads and $C_{\text{out}} = H C_h$. Let $f(a) = Va$ be a pointwise "value map" and let $g : \mathbb{R}^{C_{\text{in}}} \to (\mathbb{R}^d)^H$ be a pointwise displacement map with components $g^{(h)}$. For $u \in \mathcal{F}^{C_{\text{in}}}(\Omega)$ define $v := f[u]$ and split it into heads

$$v = v^{(1)} \oplus \cdots \oplus v^{(H)} \quad \text{with} \quad v^{(h)} \in \mathcal{F}^{C_h}(\Omega).$$

We then

(a) predict the $H$ displacement fields,

$$(\varrho^{(1)}, \ldots, \varrho^{(H)}) := g[u]$$

We emphasize that (i) each $\varrho^{(h)}$ may depend on *all* input channels (not only channels in head $h$); and (ii) locality is preserved, i.e., $\varrho^{(h)}(x)$ only depends on $u(x)$ but not $u(x')$ for $x' \neq x$.

(b) warp, or pull back the input features along the computed displacements,

$$w^{(h)} := (\text{Id} + \varrho^{(h)})^* v^{(h)}$$

(c) compute output by concatenating the heads

$$\text{SELFWARP}_{V,g}[u] := w^{(1)} \oplus \cdots \oplus w^{(H)}.$$

All nonlocality enters through evaluating $v^{(h)}$ at $x + \varrho^{(h)}(x)$. In practice $u$ is sampled on a discrete grid and $\varrho^{(h)}(x)$ generally points to fractional coordinates. We realize $(\text{Id} + \varrho^{(h)})^* v^{(h)}$ by reading $v^{(h)}$ at the displaced coordinates via bilinear interpolation; this is the same differentiable sampling function used by spatial transformer networks (Jaderberg et al., 2015) and deformable convolutions (Dai et al., 2017; Zhu et al., 2021).

**Residual pullback block (flower block)**  We embed the multihead SELFWARP in a residual block. Given an input $u$ we first compute a warped update $\text{SELFWARP}_{V,g}[u]$ and add a skip connection via a $1 \times 1$ projection,

$$u \mapsto \phi \circ \text{Norm} \circ (\text{SELFWARP}_{V,g}[u] + \text{IdProj})[u]$$

where IdProj is a $1 \times 1$ convolution, $\phi$ is GELU, and Norm is a GroupNorm.

**Multiscale composition**  Simply composing these residual blocks already yields a strong backbone across a range of forecasting tasks (see Table 9 in App. A), outperforming baselines and underscoring that learned warps alone are a sufficient primitive for effective PDE solvers. To facilitate modeling multiscale and long-range interactions present in

complex dynamics we additionally scaffold FlowerBlocks in a U-Net-like multiscale structure. This approach shares the hierarchical structure of multiscale vision transformers (Liu et al., 2021) while resembling the classical multigrid V-cycle (Brandt, 1977; Briggs et al., 2000; He & Xu, 2019). Along these lines, the learned routing may resemble the "strength of connection" heuristics found in Algebraic Multigrid (AMG) methods (Ruge & Stüben, 1987) (as opposed to fixed coarsening and refinement operators). The details of our multiscale construction are laid out in Appendix C. As we show, it leads to the best empirical performance.

# 3. Theoretical Motivation

Why are FLOWERS a natural architecture for learning solution operators of time-dependent PDEs, especially those governing flows and waves? We sketch three complementary perspectives that connect the architecture to classical structure in PDEs and numerical schemes. These perspectives are primarily motivational: we do not claim that any one of them causally explains the architecture's empirical success, but together they provide a coherent intuition and suggest routes towards a more rigorous theory. Our hope is that the reader will find at least one of the three vignettes illuminating.

## 3.1. Scalar conservation laws

Consider a scalar conservation law for $u$,

$$\begin{cases} \partial_t u(t, x) + \nabla_x \cdot (G(u(t, x))) = 0, \quad u(0, x) = u^0(x), \\ u : \mathbb{R}_+ \times \mathbb{R}^d \to \mathbb{R}, \qquad G : \mathbb{R} \to \mathbb{R}^d. \end{cases}$$

Canonical examples include the Burgers equation and traffic-flow models. Since $G$ depends on $x$ only through $u(t, x)$, the chain rule yields the equivalent quasilinear transport form, sometimes called a nonlinear optical flow (Aubert & Kornprobst, 2006),

$$\partial_t u(t, x) + A(u(t, x)) \cdot \nabla_x u(t, x) = 0$$

where $A(u) = \frac{d}{du} G(u)$. This lets us introduce characteristics: define $\Phi_t(x)$ by

$$\partial_t \Phi_t(x) = A(u(t, \Phi_t(x))), \quad \Phi_0(x) = x.$$

As long as a classical solution exists (i.e., prior to shock formation), $u$ is constant along characteristics,

$$u(t, \Phi_t(x)) = u(0, x).$$

Equivalently,

$$u(t, x) = (\Phi_t^{-1})^* u^0(x).$$

More generally, writing $\Phi_{t \leftarrow s} \stackrel{\text{def}}{=} \Phi_t \circ \Phi_s^{-1}$ for the flow map from time $s$ to $t$, we have

$$u(t, \cdot) = (\Phi_{t \leftarrow s}^{-1})^* u(s, \cdot).$$

The relevance to FLOWERS is immediate: locally in time, the solution operator is a pullback of the state by a warping map $\Phi_{t \leftarrow s}^{-1}$. More can be said. Over a short step $\Delta t$ we have the expansion

$$\Phi_{t+\Delta t \leftarrow t}^{-1}(x) = x - \Delta t A(u(t, x)) + O(\Delta t^2)$$

so to first order the displacement needed to compute $u(t + \Delta t, x)$ only depends on the *local* state $u(t, x)$, not on values $u(t, x')$ at other locations $x' \neq x$. This is precisely what our SELFWARP primitive does. A detailed derivation in multiple dimensions is given in Appendix B.

Figure 6 provides empirical evidence: on the shear flow dataset, the learned displacements visibly align with the underlying fluid velocity, suggesting that FLOWER discovers physically meaningful transport directions. From a deep learning engineering point of view, the locality of the flow map results in parameter and compute efficiency.

### 3.1.1. GENERALIZATIONS

Generalizing these ideas to systems of conservation laws—especially those with constraints and additional structure, as in compressible Euler and Navier–Stokes—is more delicate. One example is that non-hyperbolic effects like divergence-free projectors may require convolutional operators. Interestingly, warps can approximate general integral operators. Consider the non-adaptive special case in which the offsets are fixed, $\varrho^{(h)}(x) \equiv -h$, rather than predicted from the input. A head then returns the shifted field $u(x - h)$. If such heads are combined with scalar or matrix-valued weights $a(x, h)$, the continuum-head limit gives

$$(T_a u)(x) = \int_{\mathbb{R}^d} a(x, h) \, u(x - h) \, dh.$$

Changing variables $y = x - h$ yields

$$(T_a u)(x) = \int_{\mathbb{R}^d} a(x, x - y) \, u(y) \, dy,$$

which is an integral operator with kernel $K(x, y) = a(x, x - y)$. If $a$ is independent of $x$, this reduces to a translation-invariant convolution. A finite multihead warp can be interpreted as the quadrature rule,

$$(T_{a,H} u)(x) = \sum_{r=1}^{H} \omega_r \, a(x, h_r) \, u(x - h_r).$$

Thus fixed-offset warps recover the usual convolutional and kernel-operator primitives as non-adaptive special cases, but the central point is that FLOWER makes the sampling locations state dependent. A transport update such as a characteristic flow is represented directly by moving the query location, whereas a fixed convolutional stencil would

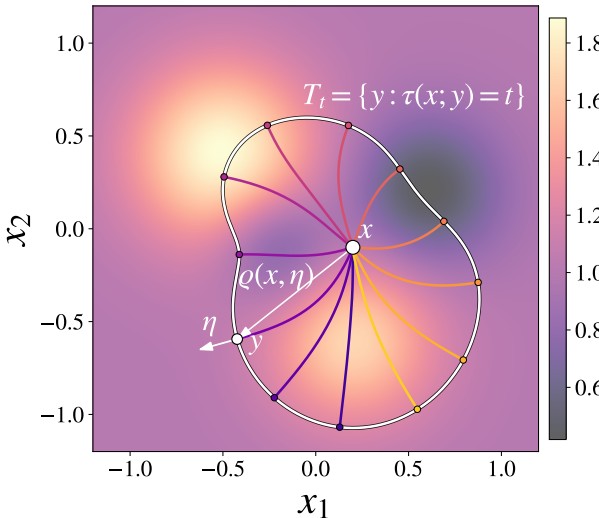

*Figure 2.* Wavefronts and rays picture: at high frequency, localized wave packets propagate along rays (see Appendix E). The solution at $(t, x)$ is a sum of contributions arriving along rays indexed by the angular variable $\eta$, which FLOWERS represent by heads: each head implements one upstream pullback at the displaced source coordinate.

have to approximate the same moving Dirac mass by a large set of fixed samples.

In the next section we outline a different, wave-related perspective which requires multiple heads.

### 3.2. Waves and geometric optics

Consider the variable-speed wave equation

$$\partial_t^2 u - c(x)^2 \Delta u = 0,$$

with Cauchy data $u(0, \cdot) = u_0$, $\partial_t u(0, \cdot) = u_1$. In the geometric-optics regime (and away from caustics), the solution at $(t, x)$ is well approximated by a superposition of contributions arriving along rays. Equivalently, writing $\Gamma_{-t}(x, \eta)$ for the point reached by tracing a ray *backwards* from $x$ for time $t$ with initial direction $\eta \in \mathbb{S}^{d-1}$, one obtains the pullback representation

$$u(t, x) \approx \int_{\mathbb{S}^{d-1}} \widehat{W}_0(t, x, \eta)\, u_0(\Gamma_{-t}(x, \eta))\, d\eta \\ + \int_{\mathbb{S}^{d-1}} \widehat{W}_1(t, x, \eta)\, u_1(\Gamma_{-t}(x, \eta))\, d\eta. \tag{1}$$

for smooth weights $\widehat{W}_{0,1}$ (absorbing Jacobian/amplitude factors); Appendix E derives this standard parametrix from a Green's function/WKB ansatz.

Equation (1) matches a *multihead pullback* layer: each head samples the input at a displaced coordinate, and pointwise channel mixing supplies the weights. Approximating the

angular integral by a learned quadrature yields

$$\sum_{h=1}^{H} \alpha_{0,1}^{(h)}(t, x)\, u_{0,1}(\Gamma_{-t}(x, \eta^{(h)}(t, x))), \tag{2}$$

i.e., a sum of pullbacks. In FLOWERS, both the effective directions $\eta^{(h)}$ ($\equiv$ displacements) and weights $\alpha^{(h)}$ are produced *pointwise* from local lifted features at $x$ (state, coefficients/geometry channels, coordinates, and metadata such as $\Delta t$), yielding an adaptive alternative to uniform angular discretization. Multiple heads also naturally represent multiple arrivals/branches, with subsequent local mixing modeling interference and mode coupling.

### 3.3. A kinetic theory perspective

Our third perspective is the natural continuum limit of FLOWERS. Treating the discrete head index as a sampling of a continuous phase variable $\eta$ and a deep cascade of FlowerBlocks as a forward-Euler discretization in time, one obtains a generalized kinetic transport equation in the lifted phase space $(x, \eta)$,

$$\partial_t u(t, x, \eta) + \mathcal{B}\big(\eta; u(t, x, \cdot)\big) \cdot \nabla_x u(t, x, \eta) \\ = \mathcal{Q}_t\big(\eta; u(t, x, \cdot)\big), \tag{3}$$

in which $\mathcal{B}$ acts as a streaming velocity and $\mathcal{Q}_t$ as an interaction (or collision-like) term capturing local channel mixing and head coupling. The key departure from classical Boltzmann/Vlasov models is that $\mathcal{B}$ is *self-adaptive*: it is predicted from the lifted state $u(t, x, \cdot)$ rather than fixed a priori. Heads thus admit a clean interpretation as a phase-space lift indexing transport modes, with macroscopic quantities recovered as moments over $\eta$. This places FLOWERS on the same conceptual footing as kinetic numerical schemes such as lattice-Boltzmann (Krüger et al., 2017; Benitez et al., 2025), whose stream-and-collide updates have an analogous structure, and connects to the broader principle that kinetic equations serve as "master" descriptions from which macroscopic models (e.g. hydrodynamics) emerge via moments and closures. The full derivation is given in Appendix D.

## 4. Results

In this section we evaluate FLOWER on a diverse suite of operator-learning benchmarks drawn from The Well (Ohana et al., 2024) and additional collections, and compare against strong baselines. Table 1 is the main quantitative comparison; Figures 3–5 give representative qualitative comparisons, including next-step predictions in challenging rheology (viscoelastic instability), autoregressive rollouts, and 3D buoyancy-driven dynamics (Rayleigh–Taylor).

We organize the experiments as follows:

- Section 4.1 benchmarks 15–20M parameter models on

| Dataset | Next-step | | | | 1:20 Rollout | | | | 21:60 Rollout | | | |
|---|---|---|---|---|---|---|---|---|---|---|---|---|
| | FNO | CNUNET | SCOT | FLOWER | FNO | CNUNET | SCOT | FLOWER | FNO | CNUNET | SCOT | FLOWER |
| **The Well** | | | | | | | | | | | | |
| Acoustic Maze | 0.1454 | 0.0129 | 0.0361 | **0.0064** | 0.4197 | 0.0874 | 0.1996 | **0.0489** | 0.7058 | 0.2199 | 2.0202 | **0.1347** |
| Active Matter | 0.1749 | 0.0650 | 0.1050 | **0.0249** | 3.2862 | 1.7781 | 4.1055 | **1.3905** | 1.8214 | 1.6465 | 2.0372 | **1.5251** |
| Gray-Scott | 0.0372 | 0.0188 | 0.0673 | **0.0102** | 0.9125 | 0.3059 | 0.4169 | **0.2074** | 37.044 | 5.1671 | 4.5485 | **4.1788** |
| MHD 64 | 0.3403 | 0.2062 | – | **0.1165** | 1.3007 | 0.9534 | – | **0.7580** | 1.6366 | 1.4299 | – | **1.2827** |
| Planet SWE | 0.0070 | 0.0027 | 0.0041 | **0.0007** | 0.1316 | 0.0624 | 0.0518 | **0.0187** | 0.5368 | 0.4846 | 0.2504 | **0.1235** |
| Neutron Star Merger | 0.4452 | 0.3391 | – | **0.3269** | **0.5980** | 0.6529 | – | 0.6223 | **0.6209** | 0.7938 | – | 0.7202 |
| Rayleigh-Bénard | 0.2104 | 0.2171 | 0.1863 | **0.0807** | 39.038 | 12.507 | 5.6486 | **2.1661** | 32.562 | 11.271 | 11.417 | **3.7461** |
| Rayleigh-Taylor | 0.1714 | 0.1351 | – | **0.0491** | 3.0057 | 5.3894 | – | **0.5862** | 3.3618 | 7.4448 | – | **1.4880** |
| Shear Flow | 0.0769 | 0.0594 | 0.1093 | **0.0463** | 1.1245 | 0.7632 | 0.8930 | **0.2246** | 14.821 | 7.4620 | 5.9953 | **1.2926** |
| Supernova Explosion | 0.4326 | 0.4316 | – | **0.2888** | 1.0162 | 2.2913 | – | **0.8113** | **1.6350** | 13.615 | – | 2.0709 |
| TGC | 0.2720 | 0.2113 | – | **0.1700** | 3.2583 | 2.0510 | – | **1.2636** | 3.8511 | 5.9142 | – | **3.7477** |
| TRL 2D | 0.3250 | 0.2559 | 0.3555 | **0.1930** | 1.2328 | 0.7051 | 0.8299 | **0.5491** | 6.0891 | 1.3390 | 1.7280 | **1.0516** |
| TRL 3D | 0.3261 | 0.3322 | – | **0.2073** | 0.9203 | 0.7139 | – | **0.6840** | 1.9969 | **0.7885** | – | 0.9881 |
| Viscoelastic fluid[†] | 0.1914 | 0.1623 | 0.2017 | **0.0624** | 0.4284 | 0.3592 | 0.4890 | **0.3465** | – | – | – | – |
| **PDEGym** | | | | | | | | | | | | |
| CE-RM | 0.1450 | 0.1465 | 0.1981 | **0.0805** | 0.2390 | 0.2908 | 0.2754 | **0.2003** | – | – | – | – |
| Wave Layer | 0.0129 | 0.0038 | 0.0074 | **0.0024** | 0.0802 | 0.0358 | 0.0500 | **0.0194** | – | – | – | – |
| **PDEBench** | | | | | | | | | | | | |
| Diffusion-Reaction | 0.0191 | 0.0033 | 0.0150 | **0.0015** | 0.7563 | 0.0279 | 0.0949 | **0.0241** | 2.3942 | **0.0228** | 0.0959 | 0.0247 |
| Shallow Water | 0.0019 | 0.0044 | 0.0187 | **0.0010** | 0.2499 | 0.1427 | 0.1164 | **0.0076** | 0.6776 | 2.6752 | 0.8386 | **0.0187** |

*Table 1.* VRMSE for unconditioned 4→1 next-step prediction, 1:20 rollout, and 21:60 rollout. Best in bold, second best colored green. All models scaled to 15M-20M parameters, see Appendix A.2 for exact counts. En-dash (–) indicates that the model does not support 3D data (SCOT), or that the trajectory is too short to calculate 21:60 rollouts (PDEGym datasets and Viscoelastic instability). [†]Using a fixed Viscoelastic Instability dataset, see A.3.

unconditioned $4 \rightarrow 1$ next-step prediction, following the evaluation protocol of The Well.

- Section 4.2 studies scaling to larger networks and compares against a recent very large foundation model for PDEs (Herde et al., 2024).

- Section 4.3 ablates key architectural choices in FLOWERS.

Additional results are deferred to the appendix, including a conditioned $1 \rightarrow 1$ task with simulation parameters provided separately (Table 8) and a time-independent PDE benchmark (Table 4, Figure 12).

### 4.1. 4-to-1 Benchmarking

We start with The Well benchmark, where the goal is to predict the state at time index $n$ from the previous four frames $u_{n-4}, \ldots, u_{n-1}$. A key difficulty is that each dataset contains trajectories from the *same* underlying PDE family but with varying simulation parameters (e.g., coefficients, boundary or forcing settings). These parameters are *not* provided to the model and must therefore be inferred implicitly from the short history window, making the task both a forecasting and a latent system-identification prob-

lem (Buitrago Ruiz et al., 2025).

Table 1 summarizes next-step prediction performance across 18 datasets with diverse physics. FLOWER achieves the lowest error overall and is typically best by a substantial margin. This in particular indicates robustness to parameter variation within each PDE family.

We next evaluate long-horizon behavior via autoregressive rollouts: we iteratively apply the one-step predictor to generate extended trajectories. The rollout results (second block of columns in Table 1) largely mirror the one-step ranking: FLOWER remains best on most datasets, with the main exception of the neutron star merger aftermath simulation, where it is slightly outperformed by FNO.

Finally, these accuracy gains come with high practical throughput. As shown in Figure 8, FLOWER matches or exceeds the throughput of FNO and SCOT across resolutions, while substantially outperforming CNEXTU-NET, especially in 3D.

## 4.2. Scaling FLOWERS

We next study how FLOWERS scale on the Euler multi quadrants (periodic) dataset from The Well. Beyond the 17M-parameter model used throughout the main benchmark (FLOWER-Tiny), we train two larger variants: FLOWER-Small (70M parameters) and FLOWER-Medium (156M parameters). All models are trained from scratch for 40 epochs; in wall-clock terms this corresponds to roughly a week of training on 1, 2, and 4 H200 GPUs, respectively. We compare with POSEIDON-L (Herde et al., 2024), a large foundation model based on the SCOT architecture, pre-trained on a diverse collection of fluid-dynamics problems. This model was similarly trained for 40 epochs on 4 H200s.

Table 2 reports the resulting errors. Performance improves smoothly with model size, indicating that FLOWERS indeed benefit from the additional capacity, without a change in training recipe or architecture class. We also note that even a tiny FLOWER delivers results comparable to a large attention-based model.

## 4.3. Ablations

We ablate the various components of FLOWER on the viscoelastic instability dataset in order to assess their relative importance. Table 3 reports one-step prediction error and 1:20 autoregressive rollout error (VRMSE), with variants sorted by rollout performance.

It is clear that warping is essential. Removing it leaves a U-Net-like multiscale backbone without a spatial interaction mechanism, and error increases dramatically. Interestingly, a *single-head* warp already recovers much of the gain for one-step prediction; indeed it is competitive with (and here exceeds) CNUNET. This is consistent with the conservation-law viewpoint in Appendix B: a single locally predicted deformation can capture a substantial part of a transport-dominated update. What is essential is that these displacements are *learned*: replacing them with non-learned, randomly initialized warps degrades one-step prediction below even the single-head learned variant (0.2190 vs. 0.1241). Conversely, even without the U-net scaffold, a single-scale FLOWER operating at fixed resolution remains competitive with baselines (Table 9).

For long-horizon rollouts, however, a single head is not sufficient: it yields poor 1:20 error even when one-step performance is reasonable. Multiple heads are therefore important for stability and accuracy over repeated composition, likely because they allow the model to represent several concurrently active transport modes rather than forcing a single displacement field to explain all motion.

Other components have smaller but consistent effects. Coordinate encodings in the lifted representation improve both one-step and rollout performance, indicating that geometric

| Model | Parameters | 1-Step | 1:20 Rollout |
|---|---|---|---|
| FLOWER-Tiny | 17.3M | 0.0160 | 0.1114 |
| FLOWER-Small | 69.3M | 0.0124 | 0.0850 |
| FLOWER-Medium | 155.8M | 0.0108 | 0.0739 |
| POSEIDON-L | 628.6M | 0.0194 | 0.1114 |

*Table 2.* Model scaling study on a compressible Euler equations dataset, loss given as VRMSE.

| Model Variant | 1-Step | 1:20 Rollout |
|---|---|---|
| w/o warping | 0.4974 | 0.6911 |
| single-head flow | 0.1241 | 0.6719 |
| non-learned warps | 0.2190 | 0.4138 |
| w/o coord encoding in lift | 0.0638 | 0.4065 |
| single conv flow (no MLP) | 0.0834 | 0.3998 |
| w/o IdProj | 0.0712 | 0.3816 |
| w/o GroupNorm | **0.0622** | 0.3760 |
| **Full Model** | 0.0624 | **0.3465** |
| vs. CNUNET | 0.1623 | 0.3592 |

*Table 3.* Ablation study on FLOWER components, evaluated on the viscoelastic instability dataset (via VRMSE loss), sorted by 1:20 rollout loss. CNUNET values given for comparison.

context is useful even when displacements are predicted pointwise. Replacing the displacement MLP with a single convolution degrades results, suggesting that the headwise motion benefits from richer pointwise nonlinearities. Finally, the residual projection (IdProj) and normalization mainly affect rollouts: removing IdProj or GroupNorm typically worsens 1:20 performance, even when the one-step metric changes only modestly. Notably, omitting GroupNorm slightly improves one-step error while harming rollouts, consistent with normalization primarily stabilizing repeated composition.

We emphasize that the relative importance of these components depends on the problem. For example, coordinate encodings may matter more with complex boundaries or discontinuous coefficients. Likewise, multiple heads should matter more when several transport families are active (e.g., multi-field coupling, mixed wave types, or strongly heterogeneous media).

## 5. Discussion

Our experimental results show that multihead warping is a strong primitive for PDE learning on a broad range of phenomena. It yields lower error in forecasting complex dynamics than baselines of comparable size, and sometimes than much larger and much more resourced models. It also scales well and admits high throughput at high resolution,

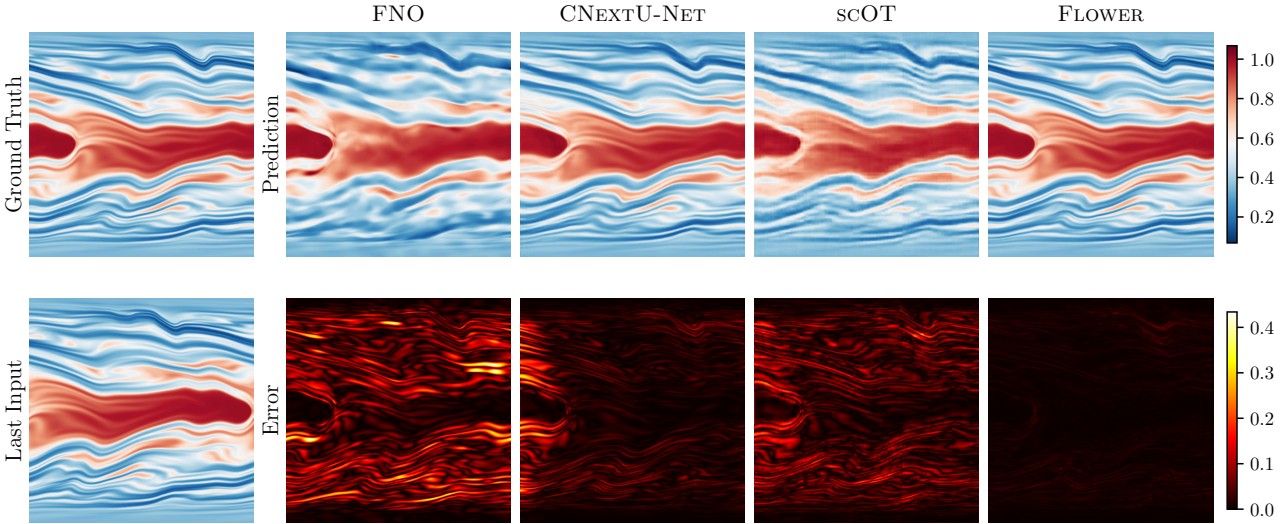

*Figure 3.* Comparison of model predictions on the viscoelastic instability dataset (4→1 unconditioned setting), showing the conformation tensor entry $C_{zz}$. The top row shows the ground truth and the prediction of each model, bottom row shows the last of the four input frames and prediction errors.

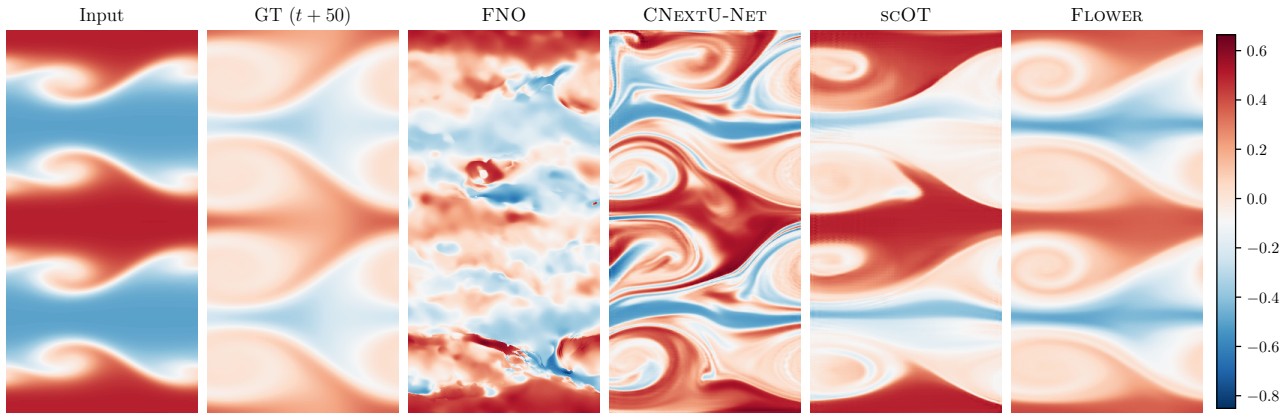

*Figure 4.* Comparison of autoregressive rollout model predictions on the shear flow dataset (4→1 unconditioned setting), showing the tracer.

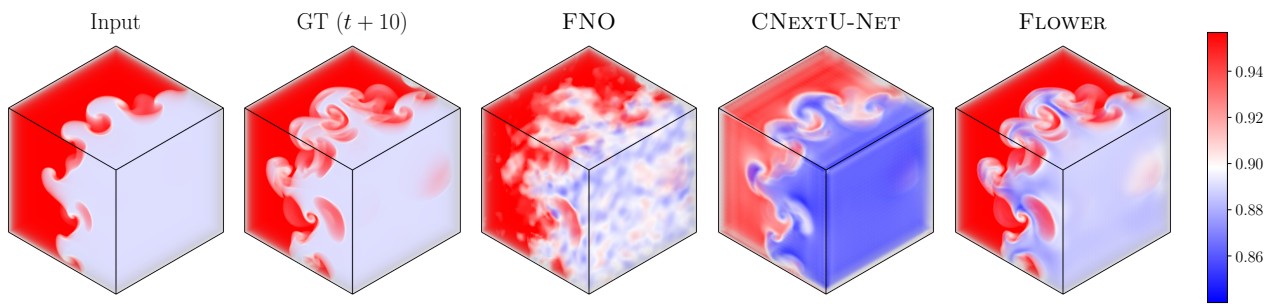

*Figure 5.* Comparison of model predictions on the Rayleigh-Taylor instability dataset (4→1 unconditioned setting), showing the density. Visualized using vape4d (Koehler et al., 2024).

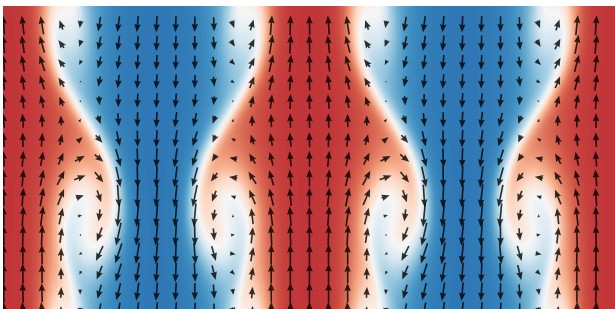

*Figure 6.* Visualization of a learned SELFWARP displacement field for the Shear Flow dataset (cf. Fig. 4). The black arrows show the learned displacement field $\varrho^{(1)}(x)$ of the first head in FLOWER's first downsampling block, overlaid on the tracer field. Without explicit supervision, the model learns displacements that align remarkably well with the underlying fluid velocity, providing evidence that the strong performance of FLOWER comes from discovering physically-meaningful transport directions. For more visualizations, see Fig. 13.

even in 3D, which is often challenging.

### 5.1. Limitations

What we currently understand well is why warps are a natural mechanism for waves and flows: characteristics for conservation laws and rays in geometric optics both imply pullback-like updates, and the continuum limit offers a kinetic view in which heads index transport modes. The same picture also suggests, thu=oughless rigorously, why FLOWERS handle diffusive and time-harmonic problems. What we do not yet understand is when the learned warps align with physically meaningful transports (and when they do not), and how best to stabilize rollouts, where FLOWER achieves the strongest results but with clear room for improvement.

On diffusive problems like Gray–Scott and active matter, FLOWERS outperform strong baselines of comparable size but do not match Poseidon (Herde et al., 2024) and Walrus (McCabe et al., 2025). One reading of why warps reach diffusive physics at all comes from the kinetic limit (3): the head-coupled interaction term $\mathcal{Q}_t$ plays the role of a collision operator, and Boltzmann-type kinetic equations are master models from which dissipative hydrodynamics emerges by taking moments. By contrast, the corresponding continuum limit of self-attention is a *collisionless* mean-field/Vlasov flow with streaming alone (Geshkovski et al., 2025; Furuya et al., 2025), structurally well matched to transport but providing no obvious moment closure for dissipation. The presence of $\mathcal{Q}_t$ in FLOWERS may therefore be what lets a warp-based architecture represent diffusive dynamics at all, even if the gap to dedicated foundation models indicates the mechanism is not fully exercised by current training.

Strong performance on the time-independent anisotropic Helmholtz benchmark (Table 4) is another point that calls for a deeper investigation. Time-harmonic solutions of the wave equation are the temporal Fourier transform of wave-equation solutions, and at high frequency the Helmholtz Green's function admits a WKB ansatz $G_\omega(x; y) \approx A(x; y)\, e^{i\omega\tau(x;y)}$ with the same eikonal travel time $\tau$ and amplitude $A$ as in Appendix E. Convolving against the source representation, the solution at $x$ is, up to dispersive phases, a superposition of source values along rays indexed by an angular variable $\eta$, i.e., a sum of pullbacks. The frequency variable replaces time, but the ray-and-warp structure is preserved, consistent with the multi-head SELFWARP primitive being effective frequency-by-frequency. This is one possible entry point for a more rigorous explanation.

These arguments are heuristic: we do not yet have evidence that learned heads track physical (or, more precisely, phase-space) velocity modes on diffusive problems or ray directions at Helmholtz frequencies. They do, however, suggest why the warp-and-mix primitive is not a priori restricted to transport-dominated regimes, and they point to concrete diagnostics (moment statistics over heads, angular spectra of $\varrho^{(h)}$) for future work.

The architecture is effective across all benchmarks considered; however, several practical limitations remain. First, boundary conditions are handled by the sampler used to evaluate $u$ at the displaced coordinates $x + \varrho^{(h)}(x)$: on periodic domains we wrap around, and on non-periodic domains queries outside simply return zero. A combination of periodic/non-periodic is supported by the code. Empirically, we found this approach to work well, but principled handling of more complex boundary conditions could be advantageous. Second, the architecture as presented assumes a structured grid. Extending FLOWERS to unstructured meshes is feasible as displacements and sampling generalize naturally to neighbor-graph or point-cloud formulations, but presents an engineering challenge which is out of scope for this paper.

## Acknowledgements

TM and ID were partially supported by the European Research Council Consolidator Grant 101232533 (PhaseShift). ML was partially supported by the Advanced Grant project 101097198 of the European Research Council, Centre of Excellence of Research Council of Finland (grant 336786) and the FAME flagship of the Research Council of Finland (grant 359186). The views and opinions expressed are those of the authors only and do not necessarily reflect those of the funding agencies or the EU. MVdH gratefully acknowledges the support of the Department of Energy, BES program under grant DE-SC0020345, Oxy, the corporate members of

the Geo-Mathematical Imaging Group at Rice University and the Simons Foundation under the MATH+X Program. TM would like to thank Lilian Gasser for the considerable time and expertise she dedicated to the design of Figure 1.

## Impact Statement

This paper presents work whose goal is to advance the field of Machine Learning. There are many potential societal consequences of our work, none which we feel must be specifically highlighted here.

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

|  | FNO | CNextU-net | sCOT | Flower |
|---|---|---|---|---|
| Optimal LR | $1 \times 10^{-3}$ | $1 \times 10^{-3}$ | $5 \times 10^{-4}$ | $1 \times 10^{-3}$ |
| VRMSE | 0.0987 | 0.1425 | 0.3068 | **0.0463** |

*Table 4.* Predictions on the 15Hz acoustic time-harmonic Helmholtz equation problem from WaveBench.

## Map of Appendices

- Appendix A contains detailed information about the experiments, models and datasets, as well as additional experimental results, including the time-independent Helmholtz equation (wave speed to solution map) where Flower surprisingly performs well. It also compares the computational efficiency of Flowers to other architectures.

- Appendix B expands the discussion of the relationship between Flowers and conservation laws.

- Appendix **??** shows how convolutions can be recovered as a special case of our layer.

- Appendix C gives a detailed description of the Flower architecture.

- Appendix D derives the kinetic continuum limit of cascaded Flower blocks.

- Appendix E shows that geometric optics solutions to the linear wave equation naturally yield a multihead pullback structure.

- Appendix F describes related work on neural PDE solvers and neural operators.

## A. Experiment Details

### A.1. Benchmark Setup

In our experiments we generally follow the benchmark settings of The Well: we aim for a batch size utilizing above 90% of GPU memory (or the maximum possible otherwise), and limit training to 24 hours on a single NVIDIA H200 GPU (The Well uses 12 hours on an H100). This evaluates performance under a fixed wall-clock budget. As The Well contains datasets where simulation parameters varies across trajectory, we either need to provide these parameters directly to the model, or teach the model to infer them from a history of snapshots. We do the former in a 1-input-1-output setting, where the model receives simulation parameters as a separate tensor for use in conditioning FiLM layers. For the latter, we follow the benchmark setting of The Well, and provide 4 inputs to each model, stacked along the channel axis.

For the 4→1 setting depicted in Table 1, all models were tested on three different learning rates ($1 \times 10^{-3}$, $5 \times 10^{-4}$, $1 \times 10^{-4}$) and the weights of the best validation VRMSE was selected. See Table 5 for the optimal learning rate per model and problem. For the 1→1 setting, where we give simulation parameters directly to the models, we ran a limited suite of experiments with a single learning rate per model, with the results depicted in Table 8.

### A.2. Models

In this section we describe the models used for comparison. Following the experimental setup of the The Well benchmark, we scale all models to between 15 and 20 million parameters for 2D problems. We naturally extend these models to 3D (as is done in The Well), which leads to modest increases in parameter size for the CNUnet (approx. 22M parameters) and Flower (approx. 24M parameters), but causes the FNO to reach almost 300M parameters. As sCOT does not support 3D-data, we omit it from 3D benchmarks.

We train all models using AdamW (Loshchilov & Hutter, 2019) and a linear warm up of the learning rate over 5 epochs, followed by cosine decay. All models were trained for 100 epochs or until 24 hours of training time on an H200 GPU, whichever came first. This time constraint intentionally biases our comparison toward more efficient models and follows the spirit of The Well's benchmark setup. Model-specific hyperparameters (learning rate and weight decay) were set according to their respective original papers, with details provided in the following subsection.

| Collection | Dataset | FNO | CNUNET | SCOT | FLOWER |
|---|---|---|---|---|---|
| The Well | `acoustic_scattering_maze` | $1 \times 10^{-3}$ | $1 \times 10^{-3}$ | $1 \times 10^{-3}$ | $1 \times 10^{-3}$ |
| | `active_matter` | $1 \times 10^{-3}$ | $1 \times 10^{-3}$ | $1 \times 10^{-3}$ | $1 \times 10^{-3}$ |
| | `gray_scott_reaction_diffusion` | $5 \times 10^{-4}$ | $1 \times 10^{-4}$ | $1 \times 10^{-3}$ | $5 \times 10^{-4}$ |
| | `MHD_64` | $1 \times 10^{-3}$ | $1 \times 10^{-3}$ | – | $5 \times 10^{-4}$ |
| | `planetswe` | $1 \times 10^{-4}$ | $1 \times 10^{-4}$ | $1 \times 10^{-3}$ | $5 \times 10^{-4}$ |
| | `post_neutron_star_merger` | $5 \times 10^{-4}$ | $1 \times 10^{-3}$ | – | $5 \times 10^{-4}$ |
| | `rayleigh_benard` | $1 \times 10^{-4}$ | $1 \times 10^{-3}$ | $5 \times 10^{-4}$ | $1 \times 10^{-3}$ |
| | `rayleigh_taylor_instability` | $5 \times 10^{-4}$ | $5 \times 10^{-4}$ | – | $1 \times 10^{-3}$ |
| | `shear_flow` | $5 \times 10^{-4}$ | $1 \times 10^{-4}$ | $1 \times 10^{-3}$ | $5 \times 10^{-4}$ |
| | `supernova_explosion_128` | $1 \times 10^{-3}$ | $5 \times 10^{-4}$ | – | $1 \times 10^{-3}$ |
| | `turbulence_gravity_cooling` | $5 \times 10^{-4}$ | $5 \times 10^{-4}$ | – | $5 \times 10^{-4}$ |
| | `turbulent_radiative_layer_2D` | $1 \times 10^{-3}$ | $5 \times 10^{-4}$ | $1 \times 10^{-3}$ | $1 \times 10^{-3}$ |
| | `turbulent_radiative_layer_3D` | $1 \times 10^{-3}$ | $1 \times 10^{-4}$ | – | $1 \times 10^{-3}$ |
| | `viscoelastic_instability` | $1 \times 10^{-3}$ | $5 \times 10^{-4}$ | $5 \times 10^{-4}$ | $1 \times 10^{-3}$ |
| PDEGym | CE-RM | $1 \times 10^{-3}$ | $1 \times 10^{-3}$ | $1 \times 10^{-3}$ | $1 \times 10^{-3}$ |
| | Wave-Layer | $1 \times 10^{-3}$ | $1 \times 10^{-3}$ | $1 \times 10^{-3}$ | $1 \times 10^{-3}$ |
| PDEBench | `diffusion_reaction` | $1 \times 10^{-4}$ | $5 \times 10^{-4}$ | $1 \times 10^{-3}$ | $1 \times 10^{-3}$ |
| | `shallow_water` | $1 \times 10^{-3}$ | $5 \times 10^{-4}$ | $1 \times 10^{-3}$ | $1 \times 10^{-3}$ |

*Table 5.* Best learning rate 4→1 next-step prediction.

| Model | 2D Parameters |
|---|---|
| FNO | 18,929,704 |
| CNUNET | 18,570,807 |
| SCOT | 17,779,868 |
| FLOWER | 17,329,362 |

*Table 6.* Approximate parameter counts for the 2D 4→1 benchmark configurations (the exact number depends on the input/output channels of the dataset, and thus varies slightly). 3D variants grow to ∼22M (CNUNET), ∼24M (FLOWER), and ∼300M (FNO); SCOT does not support 3D.

### A.2.1. FNO

We employ a standard FNO architecture consisting of three main components: an initial lifting layer that projects inputs to a high-dimensional latent space, a series of FNO blocks that perform spectral convolutions, and a final projection layer that maps back to the desired number of output channels.

Following the `neuralops` (Kossaifi et al., 2024; Kovachki et al., 2023) implementation, we use a local 2-layer MLP for both the lifting and projection operations. The lifting layer employs an inverted bottleneck structure, where the hidden dimension of the MLP is $2\times$ the latent dimension. Similarly, each FNO block follows the `neuralops` architecture, with one exception: we insert an InstanceNorm layer before the non-linearity, which we expand to a FiLM layer in case of conditioning. Specifically, each block performs a spectral convolution restricted to low-frequency modes in parallel with a local linear skip connection. The outputs of these two operations are summed and passed through the normalization layer, then through a GELU activation function. Following POSEIDON's (Herde et al., 2024) FNO implementation, we use InstanceNorm rather than alternatives such as GroupNorm or LayerNorm.

As input, we concatenate the initial field with a positional grid and the conditioning information. We choose a lifting dimension of 180, 16 fourier modes and use 4 blocks, resulting in a total of approximately 18,929,704 parameters.

In The Well benchmark, the median optimal learning rate for the FNO was $1 \times 10^{-3}$, which we adopt here. We also copy the weight decay of $1 \times 10^{-2}$.

### A.2.2. CNEXTU-NET

We use the CNextU-Net architecture as described in the Well benchmark, with one modification: when conditioning is applied, we replace the standard LayerNorm in each ConvNeXt block with a FiLM LayerNorm layer. As input, we

concatenate the initial field with the conditioning information. We adopt the hyperparameters specified in the Well benchmark, resulting in a model with approximately 18,570,807 parameters.

In The Well benchmark, the median optimal learning rate for the CNextU-Net was $1 \times 10^{-3}$, which we adopt here. We also copy the weight decay of $1 \times 10^{-2}$.

### A.2.3. scOT

We include scOT (Herde et al., 2024) for comparison with an attention-based method. While the original scOT implementation expects square 2D domains, we extend the model to handle rectangular 2D domains. Since scOT already employs time-conditioned LayerNorm layers for lead-time prediction, we repurpose these for general conditioning.

We configure the model with a patch size of 4, embedding dimension of 32, depths of [8, 8, 8, 8], attention heads of [2, 4, 8, 16], one residual block per skip connection stage, window size of 7, and MLP ratio of 4.0, yielding a model with approximately 17,779,868 parameters.

We use a learning rate of $5 \times 10^{-4}$ and weight decay of $1 \times 10^{-6}$ as reported in the original paper (Herde et al., 2024).

### A.2.4. FLOWER

For a detailed description of the architecture we refer to Section 2 and in particular Appendix C.

FLOWER, as used in the benchmark setting, has an initial lift to 160 channels, and 4 levels encoding/decoding. We choose 40 heads and set the number of GroupNorm groups to 40. This model has approximately 17,329,362 parameters.

For our scaling analysis, we constructed versions of this model with increased channel and head numbers, scaled in proportion. FLOWER-S has an initial lift to 320 channels with 80 heads, giving approximately 69,274,725 parameters and FLOWER-M lifts to 480 channels with 120 heads, giving approximately 155,828,885 parameters.

### A.3. Datasets

We adapt the codebase from The Well (Ohana et al., 2024) and use most datasets (Mandli et al., 2016; Maddu et al., 2024; Burkhart et al., 2020; McCabe et al., 2023; Miller et al., 2019a;b; 2020; Curtis et al., 2023; Lund et al., 2024; Burns et al., 2020; Hirashima et al., 2023a;b; Fielding et al., 2020) as they are. The exception is the viscoelastic instability (Beneitez et al., 2024) dataset, which erroneously contained duplicate frames that we removed. This improves the performance of every model.

Our results on other datasets may differ from The Well's published scores due to a bug in the original benchmark code. Once corrected, we expect the results to align.

We also tested our model on two datasets from PDEBench (Takamoto et al., 2022) and a time-independent Helmholtz problem from WaveBench (Liu et al., 2024). We included the latter as an expected failure case: we did not anticipate FLOWER to perform well on this problem. Surprisingly, it does, and we do not yet fully understand why.

### A.4. More Results

#### A.4.1. DATA EFFICIENCY

Within the main 4→1 unconditioned setting, we evaluate data efficiency by training and evaluating models on subsets of the full training data ranging from 5% to 100%. This task is conducted on three different datasets drawn from The Well benchmark, selected to represent diverse physical and numerical regimes. Acoustic scattering models wave propagation through maze-like domains with highly discontinuous material properties, where sharp density contrasts and complex geometry dominate the dynamics despite otherwise simple governing equations. Rayleigh–Bénard convection captures smooth fluid dynamics with strong sensitivity to initial conditions, in which small perturbations lead to qualitatively different convection patterns and coherent structures. Supernova explosion simulations represent three-dimensional, shock-dominated compressible flows with extreme nonlinearities and large dynamic ranges arising from sudden energy injection into a turbulent medium. Together, these datasets span distinct regimes in terms of dimensionality (2D vs. 3D), regularity (smooth vs. discontinuous solutions), and governing dynamics.

Figure 7 reports VRMSE as a function of training set size for the three datasets. FLOWER consistently achieves the lowest

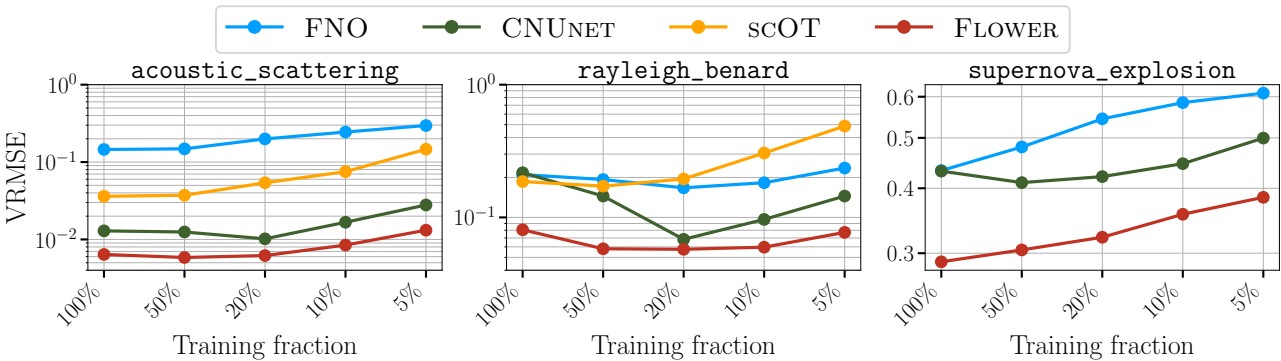

*Figure 7.* Data efficiency on the unconditioned $4 \to 1$ next-step prediction task across three datasets. Models are trained using subsets of the full training data, ranging from 5% to 100%. The x-axis shows the fraction of training data used, and the y-axis reports VRMSE. Numerical values are reported in Table 7.

| Dataset | Train fraction | FNO | CNUNET | SCOT | FLOWER |
|---|---|---|---|---|---|
| acoustic_scattering_maze | 0.05 | 0.2966 | 0.0279 | 0.1465 | 0.0132 |
| | 0.10 | 0.2443 | 0.0167 | 0.0751 | 0.0085 |
| | 0.20 | 0.1992 | 0.0102 | 0.0539 | 0.0062 |
| | 0.50 | 0.1482 | 0.0125 | 0.0372 | 0.0058 |
| | 1.00 | 0.1454 | 0.0129 | 0.0361 | 0.0064 |
| supernova_explosion_128 | 0.05 | 0.6097 | 0.4996 | – | 0.3843 |
| | 0.10 | 0.5845 | 0.4460 | – | 0.3565 |
| | 0.20 | 0.5441 | 0.4212 | – | 0.3220 |
| | 0.50 | 0.4802 | 0.4101 | – | 0.3044 |
| | 1.00 | 0.4326 | 0.4316 | – | 0.2888 |
| rayleigh_benard | 0.05 | 0.2356 | 0.1448 | 0.4878 | 0.0771 |
| | 0.10 | 0.1826 | 0.0966 | 0.3056 | 0.0597 |
| | 0.20 | 0.1671 | 0.0684 | 0.1948 | 0.0575 |
| | 0.50 | 0.1929 | 0.1449 | 0.1720 | 0.0580 |
| | 1.00 | 0.2104 | 0.2171 | 0.1863 | 0.0807 |

*Table 7.* Data efficiency in the 4→1 unconditioned next-step prediction setting. Results are reported as test score when training on subsets of the full training data.

error across all training fractions outperforming all baseline models.

## B. Motivation via Conservation Laws

FLOWERS achieve excellent performance on forecasting solutions to nonlinear systems of conservation laws. In this appendix we motivate the main primitive—a local pullback, or warping—using a simple scalar conservation law. Our goal is to build intuition, and the theory for systems is far more complicated.

In this appendix, we show that short time solutions of the scalar conservation laws can approximated using a single layer warping network. On long time intervals the shocks can appear in the solutions of the conservation laws and hence the classical solutions may not exist. However, one can define generalized viscosity solutions that exist beyond the appearance of shocks.

We consider $x \in \mathbb{R}^d$. The relevant equation is

$$\begin{cases} \partial_t u(t,x) + \nabla_x \cdot (G(u(t,x))) = 0, \quad u(0,x) = u^0(x), \\ u : \mathbb{R}_+ \times \mathbb{R}^d \to \mathbb{R}, \qquad G : \mathbb{R} \to \mathbb{R}^d. \end{cases} \tag{4}$$

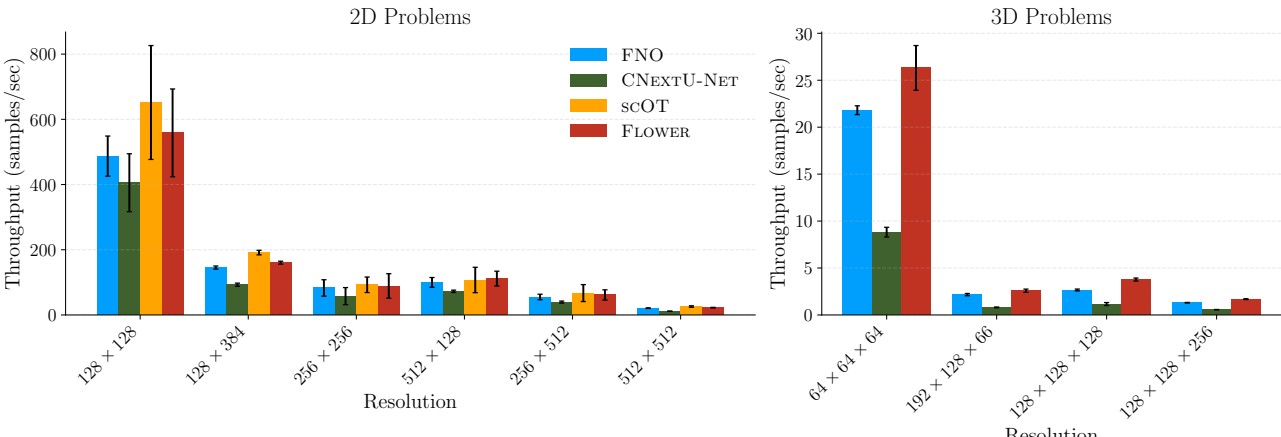

*Figure 8.* Comparison of model throughput (samples/sec) across resolutions for 2D and 3D problems, measured during the training runs of Table 1. Throughput is calculated as batch size divided by time per batch and includes both forward and backward passes. Error bars indicate standard deviation across datasets.

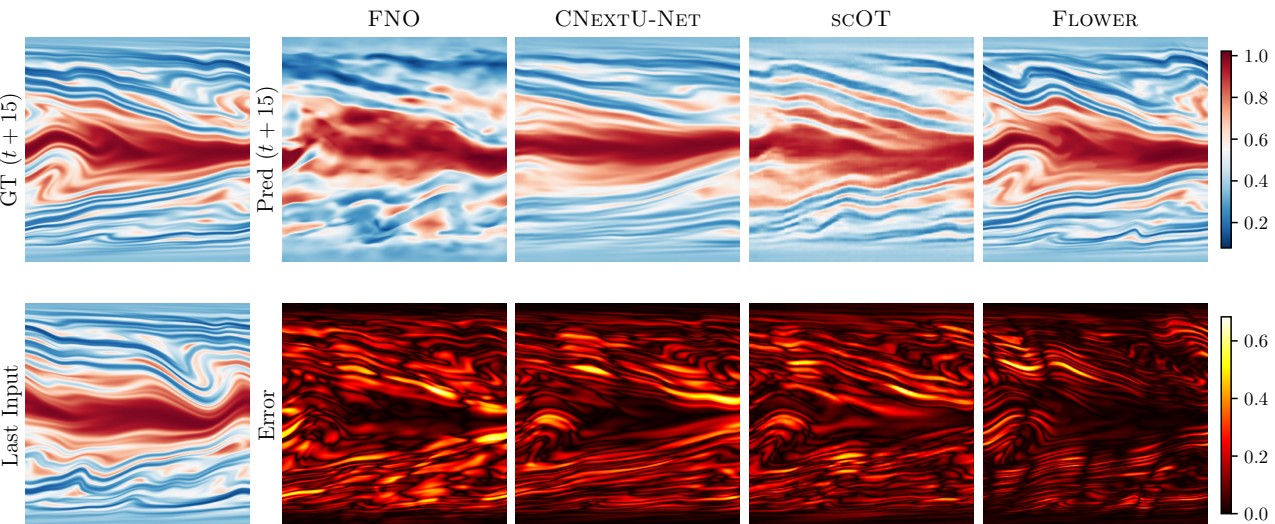

*Figure 9.* Comparison of autoregressive model predictions on the viscoelastic instability dataset (4→1 unconditioned setting), showing the conformation tensor entry $C_{zz}$. The top row shows the ground truth and the prediction of each model, bottom row shows the last of the four input frames and prediction errors.

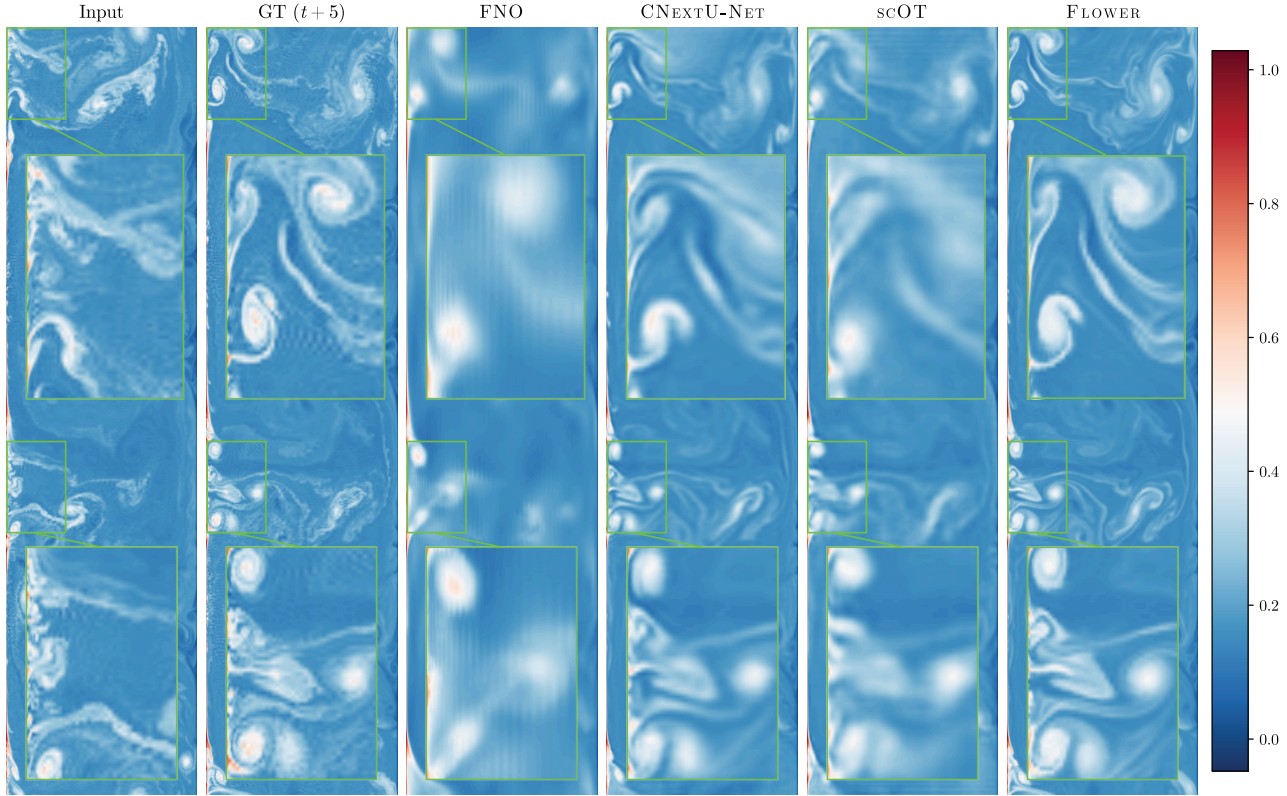

*Figure 10.* Comparison of model predictions on the Rayleigh-Bénard dataset (4→1 unconditioned setting), showing the buoyancy. We use blow-outs to enhance regions of interest.

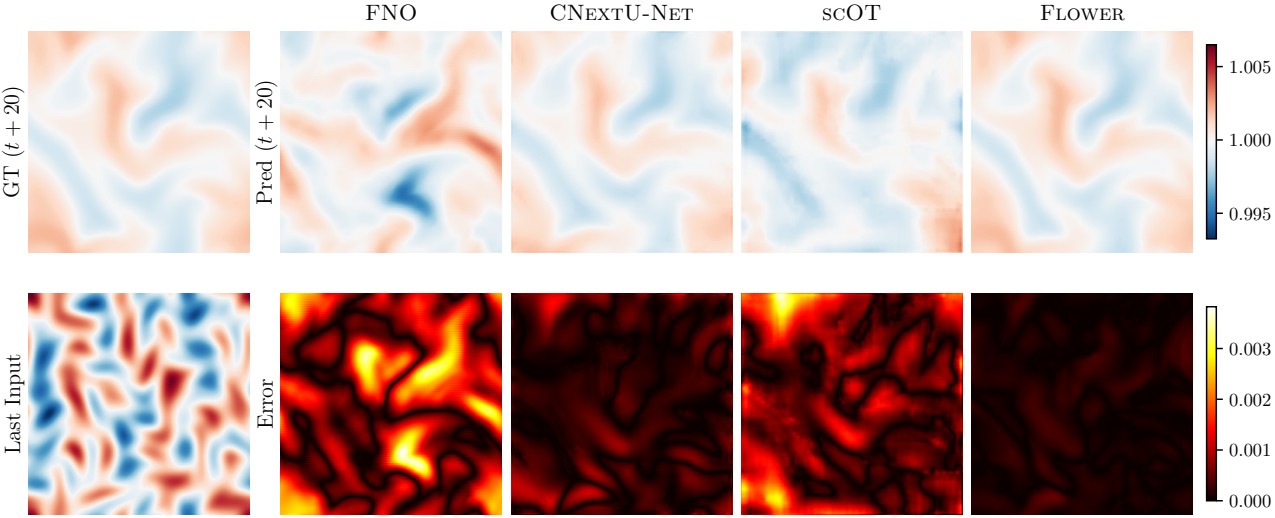

*Figure 11.* Comparison of autoregressive model predictions on the active matter dataset (4→1 unconditioned setting), showing the concentration. The top row shows the ground truth and the prediction of each model, bottom row shows the last of the four input frames and prediction errors.

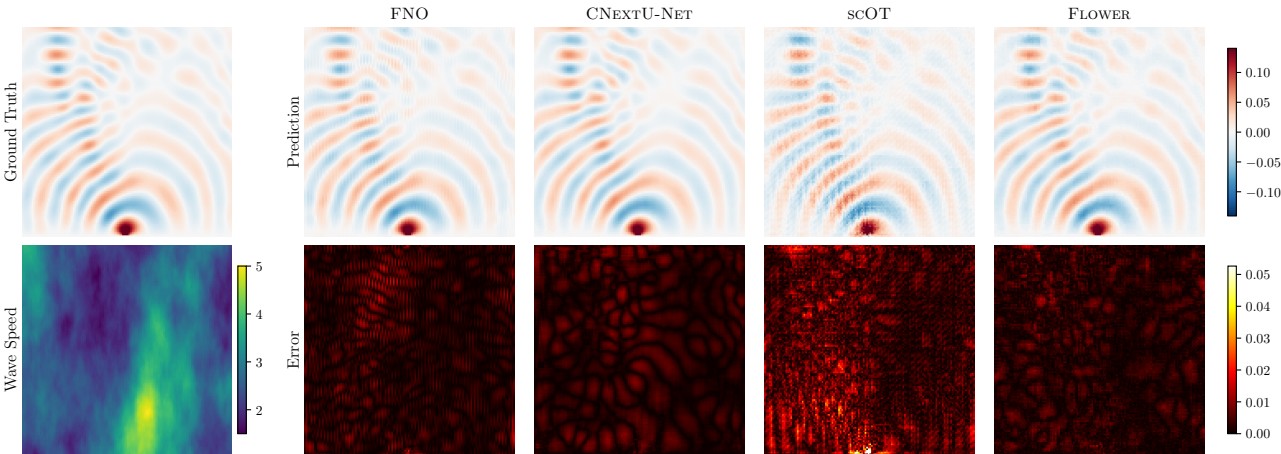

*Figure 12.* Comparison of model predictions on WaveBench's `helmholtz_anisotropic` 15Hz dataset. The top row shows the ground truth and the prediction of each model, bottom row shows the input (background wavespeed) and prediction errors.

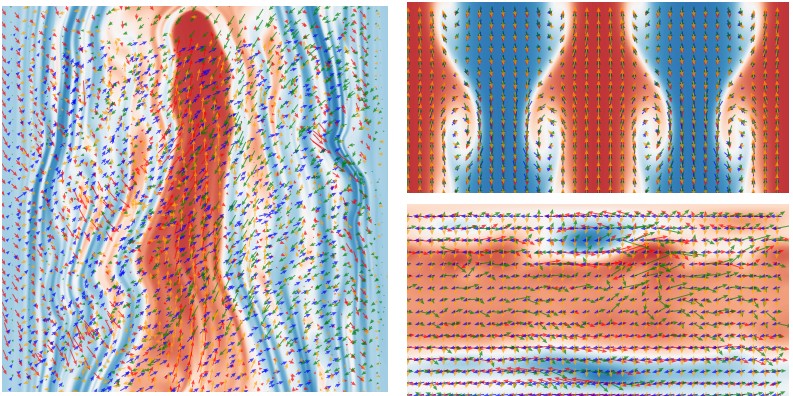

*Figure 13.* Additional visualizations of learned SELFWARP displacement fields (cf. Fig. 6). We depict the first four heads of the first downsampling block. **Left:** On the viscoelastic instability dataset, one head tracks the primary flow direction, while the remaining heads capture secondary structure. **Top right:** On shear flow, the four heads co-align with the laminar flow direction with slight variation. **Bottom right:** On Planet SWE, the displacements are more heterogeneous, reflecting the more intricate multi-physics dynamics of the rotating shallow-water system. Overall, interpretability is clearest on laminar, transport-dominated problems; on turbulent or multi-physics regimes the learned displacements remain organized but are harder to read off because there is no single coherent transport direction.

We write this as a nonlinear optical flow

$$\begin{cases} \partial_t u(t,x) + A(u(t,x)) \cdot \nabla_x u(t,x) = 0, & u(0,x) = u^0(x), \\ u : \mathbb{R}_+ \times \mathbb{R}^d \to \mathbb{R}^d, & A : \mathbb{R}^d \to \mathbb{R}^d, \end{cases} \tag{5}$$

where

$$A(u) = \frac{d}{du} G(u).$$

A generalization would be to let $A$ explicitly depend on $t$ and $x$ but we do not consider this here.

As a point of departure, characteristics are given by

$$\frac{\mathrm{d}}{\mathrm{d}t} \Phi_t(x) = A(u(t, \Phi_t(x))), \qquad \Phi_0(x) = x. \tag{6}$$

Solutions of (5) are constant along characteristic curves $t \to (t, \Phi_t(x))$. Indeed, differentiating $u(t, \Phi_t(x))$ along the

| Collection | Dataset | FNO | CNEXTU-NET | SCOT | FLOWER |
|---|---|---|---|---|---|
| The Well | `acoustic_scattering` (maze) | 0.1650 | 0.0126 | 0.0378 | **0.0077** |
| | `active_matter` | 0.2200 | 0.0918 | 0.1934 | **0.0397** |
| | `MHD_64` | 0.3543 | 0.1895 | – | **0.1378** |
| | `planetswe` | 0.0255 | 0.0041 | 0.0050 | **0.0018** |
| | `rayleigh_benard` | 0.2589 | 0.1395 | 0.2295 | **0.0706** |
| | `shear_flow` | 0.0791 | 0.0650 | 0.1383 | **0.0285** |
| | `supernova_explosion_128` | 0.5911 | 0.5038 | – | **0.3440** |
| | `turbulent_radiative_layer_2D` | 0.3326 | 0.2652 | 0.3823 | **0.1907** |

*Table 8.* VRMSE loss for conditioned $1\to1$ next-step prediction. Best results marked in bold. We did not test multiple learning rates in this setting. Instead all models were trained with a learning rate of $1 \times 10^{-3}$, except SCOT where $5 \times 10^{-4}$ was used, as reported in the original paper (Herde et al., 2024).

| | Next-step VRMSE | | | | | 1:20 Rollout VRMSE | | | | |
|---|---|---|---|---|---|---|---|---|---|---|
| Dataset | *FNO* | *CNU$_{NET}$* | *SCOT* | *FLOWER* | *FLOWER (single-scale)* | *FNO* | *CNU$_{NET}$* | *SCOT* | *FLOWER* | *FLOWER (single-scale)* |
| Active Matter | 0.1749 | 0.0650 | 0.1050 | **0.0249** | 0.0567 | 3.2862 | 1.7781 | 4.1055 | **1.3905** | 2.1708 |
| Rayleigh-Bénard | 0.2104 | 0.2171 | 0.1863 | **0.0807** | 0.1565 | 39.038 | 12.507 | 5.6486 | **2.1661** | 2.8323 |
| TRL 2D | 0.3250 | 0.2559 | 0.3555 | **0.1930** | 0.2658 | 1.2328 | 0.7051 | 0.8299 | **0.5491** | 0.6762 |
| Viscoelastic[†] | 0.1914 | 0.1623 | 0.2017 | **0.0624** | 0.0882 | 0.4284 | 0.3592 | 0.4890 | **0.3465** | 0.3921 |

*Table 9.* Results for a single-scale FLOWER variant on datasets from The Well. The model embeds inputs into $16 \times 16$ patches and processes them at fixed resolution until de-embedding, rather than using the U-net multiscale structure. Since channel counts remain constant, IdProj is unnecessary; we reallocate this convolution to a projection after the warp and rearrange operations for best performance, giving: $u \mapsto u + \phi \circ \text{Proj} \circ \text{SELFWARP}_{V,g} \circ \text{Norm}[u]$. The results in this table should be taken as a proof of concept for a single-scale FLOWER: we did not significantly iterate on the design of this architecture and we expect that much better results can be achieved if one does. Other models' results are taken from Table 1 and added for comparison. [†]Using a fixed Viscoelastic Instability dataset, see A.3.

characteristic and using equation (5),

$$
\begin{aligned}
\frac{\mathrm{d}}{\mathrm{d}t}u(t, \Phi_t(x)) &= [\partial_t u(t, z)]_{z=\Phi_t(x)} + [\nabla_z u(t, z)]_{z=\Phi_t(x)} \cdot \frac{\mathrm{d}}{\mathrm{d}t}\Phi_t(x) \\
&= [\partial_t u(t, z)]_{z=\Phi_t(x)} + [\nabla_z u(t, z)]_{z=\Phi_t(x)} \cdot [A(u(t, z))]_{z=\Phi_t(x)} \\
&= [\partial_t u(t, z) + A(u(t, z)) \cdot \nabla_z u(t, z)]_{z=\Phi_t(x)} \\
&= 0
\end{aligned}
\tag{7}
$$

and thus we see that

$$
\frac{\mathrm{d}}{\mathrm{d}t}u(t, \Phi_t(x)) = 0
$$

so that

$$
u(t, \Phi_t(x)) = u^0(x). \tag{8}
$$

Substituting (8) into (6), we find that the characteristics depend on $u(t, x)$ only through the initial condition $u_0(x)$:

$$
\frac{\mathrm{d}}{\mathrm{d}t}\Phi_t(x) = A(u^0(x)), \qquad \Phi_0(x) = x. \tag{9}
$$

As $\Phi_t(x)$ depends only on $u^0(x)$ (that is, not on values of $u^0(x')$ at points $x' \neq x$), we write it as

$$
\Phi_t(x) = \Phi_t^{u^0(x)}(x), \tag{10}
$$

and moreover,

$$\Phi_t(x) = \Phi_t^{u^0(x)}(x) = x + t\,A(u^0(x)). \tag{11}$$

This shows that (8) is equivalent to

$$u(t, x + t\,A(u^0(x))) = u^0(x). \tag{12}$$

Consider the case where $t > 0$ is so small that $t \cdot \mathrm{Lip}(A(u_0(\cdot))) < 1$. Then, by Banach fixed point theorem, the function $F_t : x \to x + tA(u^0(x))$ is invertible. Let $G_t = F_t^{-1} : \mathbb{R}^d \to \mathbb{R}^d$ be the inverse function of $F_t : \mathbb{R}^d \to \mathbb{R}^d$.

Up to this point our discussion has been precise. In what follows we will discuss, somewhat non-rigorously, why FLOWERS can be used to approximate the function $G_t$.

Let $B = A \circ u^0 : \mathbb{R}^d \to \mathbb{R}^d$. Below, we assume that $B$ is $C^3(\mathbb{R}^d)$. By using (11), we see that $G_t$ satisfies the equation

$$G_t(x) = x - tB(G_t(x)).$$

Then,

$$u(t, x) = u^0(G_t(x)). \tag{13}$$

Let us consider the Taylor expansion of $G_t(x)$ in variable $t$ around the time zero,

$$G_t(x) = x + tg_1(x) + t^2 g_2(x) + O(t^3).$$

By operating on both sides of this equation with $B$ and using Taylor-expansion around $x$,

$$B(y) = B(x) + B'(x)(y - x) + O(|y - x|^2), \quad B'(x) = DB(x) \in \mathbb{R}^{d \times d},$$

we obtain

$$B(G_t(x)) = B(x + tg_1(x) + t^2 g_2(x)) = B(x) + tB'(x)g_1(x) + O(t^2).$$

Hence,

$$x + tg_1(x) + t^2 g_2(x) = x - t(B(x) + tB'(x)g_1(x) + O(t^2)) = x - tB(x) - t^2 B'(x)g_1(x) + O(t^3).$$

Matching coefficients we get

$$g_1(x) = -B(x) = -A(u^0(x)), \qquad g_2(x) = B'(x)B(x).$$

Combining the above formulas, we obtain

$$\begin{aligned} G_t(x) &= x - tB(x) + t^2 B'(x)B(x) + O(t^3) \\ &= x - tA(u^0(x)) + t^2 D[A \circ u^0](x)\,A(u^0(x)) + O(t^3), \end{aligned}$$

where by the chain rule, the Jacobian matrix $DB(x) = D[A \circ u^0](x)$ is given by

$$D[A \circ u^0](x) = A'(u^0(x))(\nabla u^0(x))^\top,$$

where $A'$ is the derivative of the map of the map $A : \mathbb{R} \to \mathbb{R}^d$. Thus the 1st order approximation of $G_t(x)$ in $t$ is

$$G_t(x) \approx x - tA(u^0(x))$$

which defines a pointwise-dependent displacement map considered below in Section C.1. The 2nd order approximation of $G_t(x)$ in $t$ is given by

$$G_t(x) \approx x - tA(u^0(x)) + t^2 A'(u^0(x))(\nabla u^0(x))^\top A(u^0(x))$$

Here, the derivative $\nabla u^0(x)$ can approximated using a multi-head warp network (see Section C.1 below) as $\nabla u^0(x)^\top = (\frac{\partial}{\partial x^1} u^0(x), \ldots, \frac{\partial}{\partial x^d} u^0(x))$ can be approximated by

$$\frac{\partial}{\partial x^j} u^0(x) = \frac{1}{s}\left(\begin{array}{c}(Id + \varrho_{j,s})^*\\ -Id\end{array}\right)^\top \left(\begin{array}{c}u^0(x)\\ u^0(x)\end{array}\right) + O(s),$$

where $\varrho_{j,s}(x) = se_j$, $s > 0$ is small, and $e_j = (0, 0, \ldots, 0, 1, 0, \ldots, 0)$ are the unit coordinate vectors.

# C. A detailed description of the architecture

For clarity we describe the pullback layer over continuous coordinates; in practice the pullback is implemented by differentiable interpolation on a grid with a boundary extension determined by the chosen boundary conditions.

## C.1. Feature spaces and notation conventions

Let $\Omega \subset \mathbb{R}^d$ be the spatial domain, $\mathcal{F}(\Omega)$ a sufficiently rich space of real-valued functions on $\Omega$, and

$$\mathcal{F}^C := \left\{ u : \Omega \to \mathbb{R}^C \right\}$$

the space of fields with $C$ channels.

We write $u, v, w$ for fields, $x \in \Omega$ for spatial coordinates, and $a = u(x) \in \mathbb{R}^C$ for pointwise channel vectors. Operators acting on fields take arguments in brackets (e.g. $A[u]$); finite-dimensional maps take arguments in parentheses (e.g. $g(a), \tau(x)$).

**Broadcast**    Given a local map $h : \mathbb{R}^p \to \mathbb{R}^q$, its broadcast (pointwise) extension is

$$h[\,\cdot\,] : \mathcal{F}^p(\Omega) \to \mathcal{F}^q(\Omega), \qquad (h[u])(x) := h(u(x)).$$

**Concatenation**    For $u \in \mathcal{F}^{C_1}(\Omega)$ and $v \in \mathcal{F}^{C_2}(\Omega)$, we define channel concatenation as

$$u \oplus v \in \mathcal{F}^{C_1 + C_2}(\Omega), \qquad (u \oplus v)(x) = u(x) \oplus v(x) = [u_1(x), \ldots u_{C_1}(x), v_1(x), \ldots, v_{C_2}(x)].$$

**Pullback**    Given a map $\tau : \Omega \to \mathbb{R}^d$, define the associated pullback operator

$$\tau^* : \mathcal{F}^C(\Omega) \to \mathcal{F}^C(\Omega), \qquad (\tau^* v)(x) := v(\tau(x)).$$

For a displacement field $\varrho \in \mathcal{F}^d(\Omega)$ we write $\tau = \mathrm{Id} + \varrho : \Omega \to \mathbb{R}^d$ for the map $x \mapsto x + \varrho(x)$.

## C.2. Multihead SELFWARP (pullback / spatial warp)

Fix integers $C_{\mathrm{in}}, C_{\mathrm{out}} \geq 1$ and a head count $H \geq 1$ with $C_{\mathrm{out}} = HC_h$. A multihead pullback layer is determined by

- a local *value* projection $f : \mathbb{R}^{C_{\mathrm{in}}} \to \mathbb{R}^{C_{\mathrm{out}}}$ of the form $f(a) = Va$, where $V$ is a matrix;

- a local *multihead displacement* map $g : \mathbb{R}^{C_{\mathrm{in}}} \to (\mathbb{R}^d)^H$ with components $g^{(h)} : \mathbb{R}^{C_{\mathrm{in}}} \to \mathbb{R}^d$. (In the code, $g$ is a small pointwise MLP realized by $1 \times 1$ convolutions.)

We will now define the key building block operator

$$\mathrm{SELFWARP}_{V,g} : \mathcal{F}^{C_{\mathrm{in}}}(\Omega) \to \mathcal{F}^{C_{\mathrm{out}}}(\Omega).$$

For $u \in \mathcal{F}^{C_{\mathrm{in}}}(\Omega)$ set

$$v := f[u] \in \mathcal{F}^{C_{\mathrm{out}}}(\Omega), \qquad [\varrho^{(1)}, \ldots, \varrho^{(H)}] := g[u] \in \left(\mathcal{F}^d(\Omega)\right)^H,$$

split $v = v^{(1)} \oplus \cdots \oplus v^{(H)}$ with $v^{(h)} \in \mathcal{F}^{C_h}(\Omega)$, and define head warps $\tau_u^{(h)} := \mathrm{Id} + \varrho^{(h)}$. Then

$$\mathrm{SELFWARP}_{V,g}[u] = \bigoplus_{h=1}^{H} (\tau_u^{(h)})^* v^{(h)} = \bigoplus_{h=1}^{H} \left(\mathrm{Id} + g^{(h)}[u]\right)^* (f^{(h)}[u]).$$

For $H = 1$ this reduces to the single-head layer $u \mapsto (\mathrm{Id} + g[u])^* (Vu)$.

### C.3. Residual pullback block (FlowerBlock)

Fix the number of input and output channels $C_{\text{in}}, C_{\text{out}}$. A *FlowerBlock* is an operator

$$\text{Block} : \mathcal{F}^{C_{\text{in}}}(\Omega) \to \mathcal{F}^{C_{\text{out}}}(\Omega),$$

defined by

$$\text{Block} = \phi \circ \text{Norm} \circ \Big( \text{SELFWARP}_{V,g} + \text{IdProj} \Big).$$

Here $\text{IdProj} : \mathcal{F}^{C_{\text{in}}}(\Omega) \to \mathcal{F}^{C_{\text{out}}}(\Omega)$ is a pointwise linear projection (a $1 \times 1$ convolution) and $\phi$ is a pointwise nonlinearity (GELU). Norm is group normalization.[3]

### C.4. Multiscale (U-Net) Flowers

We arrange FlowerBlocks into a U-Net with $L$ levels. Let $\Omega_0$ denote the finest-resolution domain (in practice a grid), and let $\Omega_{\ell+1}$ be obtained from $\Omega_\ell$ by downsampling by a factor of 2 in each spatial direction. Define channel widths

$$c_0 := c_{\text{lift}}, \qquad c_\ell := 2^\ell c_0, \quad \ell = 0, \dots, L - 1.$$

**Positional augmentation and lifting** Let $\xi \in \mathcal{F}^d(\Omega_0)$ be the coordinate field (positional grid) so that $\xi(x) = x$. Define the augmentation operator

$$\text{Aug} : \mathcal{F}^{C_{\text{in}}}(\Omega_0) \to \mathcal{F}^{C_{\text{in}}+d}(\Omega_0), \qquad \text{Aug}[u] := u \oplus \xi,$$

and a pointwise lifting map

$$\text{Lift} : \mathcal{F}^{C_{\text{in}}+d}(\Omega_0) \to \mathcal{F}^{c_0}(\Omega_0).$$

**Encoder-side operators** For $\ell = 0, \dots, L - 2$, let

$$E_\ell : \mathcal{F}^{c_\ell}(\Omega_\ell) \to \mathcal{F}^{c_{\ell+1}}(\Omega_{\ell+1})$$

denote the encoder stage

$$E_\ell = \rho \circ \text{DownConv}_\ell \circ \text{Block}_\ell^\downarrow,$$

where $\text{Block}_\ell^\downarrow : \mathcal{F}^{c_\ell}(\Omega_\ell) \to \mathcal{F}^{c_\ell}(\Omega_\ell)$ is a FlowerBlock (same in/out channels), $\text{DownConv}_\ell$ is a stride-2 convolution mapping $c_\ell \mapsto c_{\ell+1}$, and $\rho$ is ReLU.

**Bottleneck** Let

$$\text{Bot} : \mathcal{F}^{c_{L-1}}(\Omega_{L-1}) \to \mathcal{F}^{c_{L-1}}(\Omega_{L-1})$$

be a FlowerBlock at the coarsest scale.

**Decoder-side operators** Define decoder channel sizes

$$\tilde{c}_{L-1} := c_{L-1}, \qquad \tilde{c}_\ell := 2c_\ell \ \ (\ell = 0, \dots, L - 2),$$

so that after concatenating a skip at level $\ell$ the decoder state has $2c_\ell$ channels. For $\ell = 1, \dots, L - 1$, define an up stage

$$U_\ell : \mathcal{F}^{\tilde{c}_\ell}(\Omega_\ell) \to \mathcal{F}^{c_{\ell-1}}(\Omega_{\ell-1})$$

by

$$U_\ell = \rho \circ \text{UpConv}_\ell \circ \text{Block}_\ell^\uparrow,$$

where $\text{Block}_\ell^\uparrow : \mathcal{F}^{\tilde{c}_\ell}(\Omega_\ell) \to \mathcal{F}^{c_{\ell-1}}(\Omega_\ell)$ is a FlowerBlock that reduces channels, $\text{UpConv}_\ell$ is a stride-2 transposed convolution preserving channel count $c_{\ell-1}$ while upsampling $\Omega_\ell \to \Omega_{\ell-1}$, and $\rho$ is ReLU.

---

[3]For conditioned prediction we use FiLM modulated normalization.

**Final projection** Let

$$\mathrm{Proj} : \mathcal{F}^{2c_0}(\Omega_0) \to \mathcal{F}^{C_{\mathrm{out}}}(\Omega_0)$$

be the pointwise projection

$$\mathrm{Proj} = W_2 \circ \rho \circ W_1,$$

where $W_1$ maps $2c_0 \mapsto c_0$ and $W_2$ maps $c_0 \mapsto C_{\mathrm{out}}$ (both $1 \times 1$ convolutions), and $\rho$ is ReLU.

**Definition (Flower)** The full network is an operator

$$\Phi : \mathcal{F}^{C_{\mathrm{in}}}(\Omega_0) \to \mathcal{F}^{C_{\mathrm{out}}}(\Omega_0),$$

defined by the recursion

$$x_0 := \mathrm{Lift}\big(\mathrm{Aug}[u]\big),$$

$$\text{for } \ell = 0, \ldots, L-2: \qquad s_\ell := x_\ell, \qquad x_{\ell+1} := E_\ell[x_\ell],$$

$$d_{L-1} := \mathrm{Bot}[x_{L-1}],$$

$$\text{for } \ell = L-1, \ldots, 1: \qquad \hat{d}_{\ell-1} := U_\ell[d_\ell], \qquad d_{\ell-1} := \hat{d}_{\ell-1} \oplus s_{\ell-1},$$

and finally

$$\Phi[u] = \mathrm{Proj}[d_0].$$

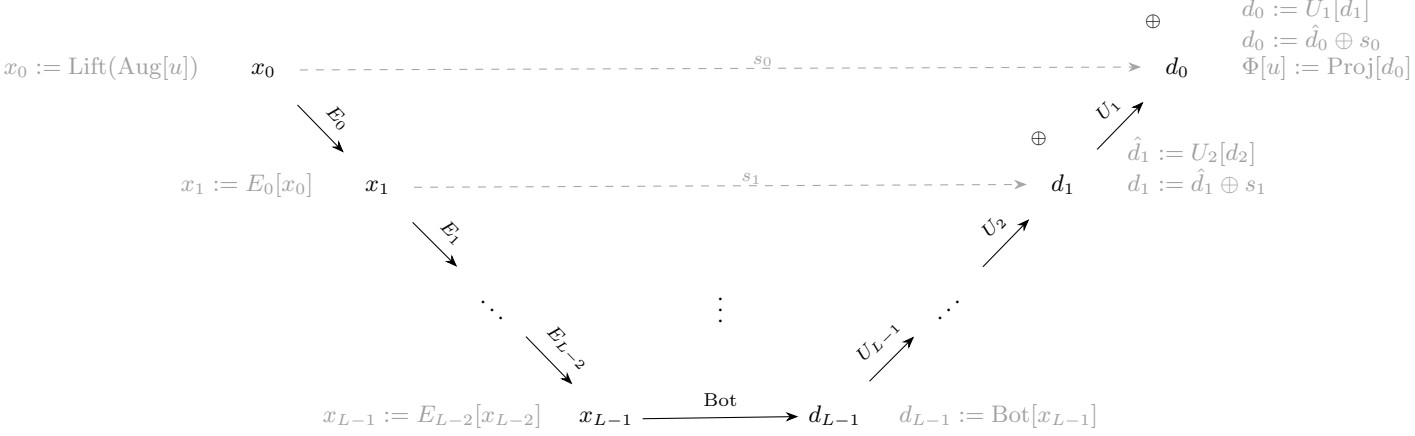

*Figure 14.* A schematic illustration of the FLOWER U-Net architecture.

## D. The continuum limit and kinetic theory perspective

We use similar notation as in Appendix C but work with a single-scale architecture, without the U-Net. In particular, let $H$ be the number of heads and write (moving the head index to the argument)

$$u(x) = (u(x,1), \ldots, u(x,H)), \quad \text{with} \quad u(x,h) \in \mathbb{R}^c,$$

so that $u \in \mathcal{F}^{Hc}(\Omega)$ and $u(x) \in \mathbb{R}^{Hc}$. A multihead SELFWARP has two pointwise components,

- a value map $V : \mathbb{R}^{Hc} \to \mathbb{R}^{Hc}$ (a $1 \times 1$ linear (affine) convolution mixing channels and heads; here we assume that we are already "in the lift" and the number of channels is constant so that we can derive a meaningful limit;

- a displacement map $g : \mathbb{R}^{Hc} \to (\mathbb{R}^d)^H$ with head components $g^{(h)} : \mathbb{R}^{Hc} \to \mathbb{R}^d$; this a pointwise MLP (implemented via a convnet with $1 \times 1$ convolutions).

For a $u \in \mathcal{F}^{Hc}(\Omega)$, let $v := V[u]$ and $\varrho^{(h)} = g^{(h)}[u]$. Define the warped outputs for each head as

$$(\text{SELFWARP}_{V,g}[u])(x, h) := (\text{Id} + \varrho^{(h)})^* v(\cdot, h)(x) = v(x + \varrho^{(h)}(x), h).$$

Note that $x \mapsto \varrho^{(h)}(x)$ only depends on $u(x)$; non-locality enters via the pullback.

We now interpret the discrete head index as a sampling of a continuous latent variable, $\eta \in \Lambda$ compact (which will turn out to have a kinetic interpretation). A mixing or integration over heads can then be viewed as an integration against some probability measure. Through a quadrature, with weights $w_h$ and samples $\eta_h$, we identify

$$u(x, \eta_h) \text{ with } u(x, h).$$

Letting $H \to \infty$, we obtain a function $u \in \mathcal{F}^c(\Omega \times \Lambda)$ that can be interpreted as a kinetic field $u(x, \eta)$, that depends on positional variable $x \in \Omega$ and a directional (velocity) variable $\eta \in \Lambda$.

A Flower layer is (approximately) computing

$$\mathcal{A}\left(\text{SELFWARP}_{V,g}[u] + P[u]\right)$$

with $P$ a pointwise linear map (a generalized skip connection) and $\mathcal{A}$ a pointwise nonlinear map.[4] We interpret this through the lens of kinetic theory: the pullback $x \mapsto \varrho(x, \eta)$ implements streaming in an $\eta$-dependent manner; the pointwise operations that mix channels and/or kinetic labels $\eta$ at a fixed $x$ implement interactions. Pointwise maps act along the kinetic variable, $\eta$, that is to say, on the functions $\eta \mapsto u(x, \eta)$ for each $x$.

**Replacing layer index by time**

By considering a probability measure $\mu$ on $\Lambda \subset \mathbb{R}^d$ and interpreting $\eta_h$ to be the quadrature nodes $\eta_h \in \Lambda$ and corresponding weights $w_k \in \mathbb{R}$, $h = 1, \ldots, H$, on the set $\Lambda$, we can consider sums over multiple heads as approximations of integrals over $\Lambda$ obtained using a quadrature formula

$$\int_\Lambda F(\eta) d\mu(\eta) \approx \sum_{h=1}^{H} w_h F(\eta_h). \tag{14}$$

We consider a composition of $N + 1$ Flower blocks indexed by $n = 0, \ldots, N$, let $\Delta t := T/N$, and write $t_n := n\Delta t$. We will interpret each layer as being "small" in the sense that we will assume that there are operators $\mathcal{B}_t(\eta; .)$ and $\mathcal{Q}(\eta; .)$ such that for each layer $n$,

- the displacement function is given by
$$\varrho(x, \eta) = \Delta t \cdot \mathcal{B}_{t_n}(\eta; u_n(x, .))$$

  where
$$\mathcal{B}_{t_n}(\eta; u_n(x, .)) = \sum_{h=1}^{H} b_{t_n}(\eta, \eta_h) u_n(x, \eta_h).$$

  In several physical applications, see e.g. (18) below, the function $\mathcal{B}_{t_n}(\eta; u_n(x, .))$ has the simple form, $\mathcal{B}_{t_n}(\eta; u_n(x, .)) = \eta$;

- the local contribution of the layer is determined by
$$P[u] = \Delta t \cdot \mathcal{Q}_{t_n}(\eta; u_n(x, .)),$$

  where
$$\mathcal{Q}_{t_n}(\eta; u_n(x, .)) = \sum_{h=1}^{H} q_{t_n}(\eta, \eta_h) u_n(x, \eta_h).$$

---

[4]GroupNorm is not strictly local in space but we ignore this here as it is not a leading order effect. It anyway globally rescales the field.

We then expand the pullback of $u_n$ as

$$u_n(x + \Delta t \cdot \mathcal{B}_{t_n}(\eta; u_n(x,.)), \eta) = u_n(x, \eta) + \Delta t \mathcal{B}_{t_n}(\eta; u_n(x,.)) \cdot \nabla_x u_n(x, \eta) + O(\Delta t^2).$$

We get

$$u_{n+1}(x, \eta) = u_n(x, \eta) + \Delta t \mathcal{B}_{t_n}(\eta; u_n(x,.)) \cdot \nabla_x u_n(x, \eta) + \Delta t \cdot \mathcal{Q}_{t_n}(\eta; u_n(x,.)). \tag{15}$$

If

$$\mathcal{A} = I + \Delta t \cdot \mathcal{A}'$$

we can account for the composition with $\mathcal{A}$ by redefining

$$\mathcal{B}_{t_n} := \mathcal{A}_{t_n} \mathcal{B}_{t_n}, \quad \mathcal{Q}_{t_n} := \mathcal{A}_{t_n} \mathcal{Q}_{t_n} + \mathcal{A}'_{t_n}. \tag{16}$$

With $t_n = n\Delta t$, through a finite difference approximation, assuming that as $N \to \infty$ and $\Delta t \to 0$ (and $\mathcal{A} \to I$), $u_n(t, x)$ converges to $u(t, x, \eta)$, we get the following PDE

$$\partial_t u(t, x, \eta) + \mathcal{B}_t(\eta; u(t, x, .)) \cdot \nabla_x u(t, x, \eta) = \mathcal{Q}_t(\eta; u(t, x, .)), \tag{17}$$

which is a nonlinear kinetic equation with a self-adaptive streaming velocity. A notable special case of this equation is

$$\partial_t u(t, x, \eta) + \eta \cdot \nabla_x u(t, x, \eta) = \mathcal{Q}_t(\eta; u(t, x, .)), \tag{18}$$

which is the familiar Boltzmann equation upon interpreting $u$ as a distribution of particles, and which is known to yield hydrodynamics by evaluating moments. The nonlinear equation (18) can be considered as a Boltzmann-like equation where the drift (the streaming velocity) depends on the local kinetic state (e.g. a distribution over microscopic velocities in the true Boltzmann case). This gives it a great deal of flexibility.

## E. Linear waves

We consider the linear wave equation in $\mathbb{R}^d$, with a variable wave speed, $c = c(x)$,

$$\partial_t^2 u - c(x)^2 \Delta u = 0, \quad (t, x) \in [0, \infty) \times \mathbb{R}^d \tag{19}$$

with initial data $u(0, x) = u_0(x)$ and $\partial_t u(0, x) = u_1(x)$. Here, we focus on the three-dimensional case, that is, $d = 3$ on such times $t$ that waves have not formed caustics, that is, the Riemannian geodesics associated to the travel time metric do not have conjugate points. We let $G(t, x; y)$ be the causal Green's function so that

$$u(t, x) = \int_{\mathbb{R}^d} \partial_t G(t, x; y) u_0(y) dy + \int_{\mathbb{R}^d} G(t, x; y) u_1(y) dy. \tag{20}$$

The derivation is standard, but we explicitly connect it with our architecture, FLOWERS. We use the geometric optics formalism. It states that for (relatively) small $t$ and away from caustics, that is, focusing of wave fronts caused by lensing effects, $G(t, x; y)$ is concentrated on the wavefront surface $t = \tau(x; y)$, where $\tau(x; y)$ is the travel time or geodesic distance from the point $y$ to the point $x$. This motivates the following "WKB Ansatz",

$$G(t, x; y) \approx A(x; y)\delta(t - \tau(x; y)), \tag{21}$$

where the amplitude coefficient $A$ encodes the so-called geometric spreading. It can be shown that this ansatz (21) is correct "up to smooth terms". What this means is that it correctly propagates the high-frequency singularities.

Inserting the ansatz (21) into the wave equation (19) yields

$$A(1 - c(x)^2|\nabla_x \tau|^2)\delta''(t - \tau) + c(x)^2(2\nabla_x A \cdot \nabla_x \tau + A\Delta_x \tau)\delta'(t - \tau) + \text{less singular terms} = 0.$$

To make $G$ solve the wave equation (19) "up to less singular terms" away from $x = y$, we set the coefficients in front of the derivatives of the Dirac's delta distributions $\delta''$ and $\delta'$ to zero (that is., we analyze the leading order wavefront singularities).

The $\delta''$ coefficient yields the well-known eikonal equation for the travel times,

$$c(x)^2|\nabla_x \tau(x; y)|^2 = 1, \quad \tau(y; y) = 0.$$

The $\delta'$ coefficient yields the transport equation which determines the geometric spreading $A(x;y)$,

$$2\nabla_x A(x;y) \cdot \nabla_x \tau(x;y) + A(x;y)\Delta_x \tau(x;y) = 0.$$

For simplicity, we focus on the $u_1$ term (and assume that $u_0 \equiv 0$ for now). Using (21),

$$\int_{\mathbb{R}^d} G(t,x;y)u_1(y)dy \approx \int_{\mathbb{R}^d} A(x;y)\delta(t - \tau(x;y))u_1(y)dy.$$

The delta function turns a volume integral into a surface integral over the surface $\{y : t(x;y) = t\}$:

$$\int_{\mathbb{R}^d} G(t,x;y)u_1(y)dy \approx \int_{t=\tau(x;y)} \frac{A(x;y)}{|\nabla_y \tau(x;y)|} u_1(y)dS(y).$$

The factor

$$W_1(x;y) := \frac{A(x;y)}{|\nabla_y \tau(x;y)|}$$

is a weight. The $u_0$ term would have the same structure, with a different weight, $W_0(x;y)$ say.

We can now make the pullback structure of the solution explicit. Before caustics form, the wavefront $\{y : \tau(x;y) = t\}$ can be parameterized by a "ray label" $\eta$, for example, the initial (tangent) direction on $\mathbb{S}^{d-1}$. Let $x(t,y,\eta)$ denote position of the ray at time $t$ that emanated at the time 0 from the point $y \in \mathbb{R}^d$ to the direction $\eta \in \mathbb{S}^{d-1}$. The rays, that are equivalent to geodesic curves determined by the Riemannian metric $c(x)^{-2}(dx_1^2 + \ldots + dx_d^2)$, are given by the ordinary differential equation,

$$\partial_t^2 x(t,y,\eta) = 2((\nabla \log c)(x(t,y,\eta)) \cdot \partial_t x(t,y,\eta))\, \partial_t x(t,y,\eta) - c(x(t,y,\eta))^2\,(\nabla \log c)(x(t,y,\eta)), \tag{22}$$

$$x(t,y,\eta)|_{t=0} = y, \ \partial_t x(t,y,\eta)|_{t=0} = \eta. \tag{23}$$

In the absence of caustics, $x$, $\eta$ and $t$ determine uniquely $y$ such that $x(t,y,\eta) = x$. We introduce the map, $\Gamma_{-t}(x,\eta) = y$. Then, under the change of variable $\eta \to \Gamma_{-t}(x,\eta)$, the surface measure on the surface $\{y \in \mathbb{R}^d : \tau(x;y) = t\}$ takes the form

$$\frac{dS(y)}{|\nabla_y \tau(x;y)|} = J(t,x,\eta)d\eta,$$

where $J$ is the relevant Jacobian. Putting everything together, we now see that we can write (20) as an integral of pullbacks

$$u(t,x) \approx \int_{\mathbb{S}^{d-1}} W_0(t,x,\eta)u_0(\Gamma_{-t}(x,\eta))\, J(t,x,\eta)d\eta + \int_{\mathbb{S}^{d-1}} W_1(t,x,\eta)u_1(\Gamma_{-t}(x,\eta))\, J(t,x,\eta)d\eta,$$

signifying pullbacks of $u_0$ and $u_1$ for given $t$ and $\eta$, or, upon introducing the "displacement", $\varrho(t,x,\eta) = \Gamma_{-t}(x,\eta) - x$,

$$u(t,x) \approx \int_{\mathbb{S}^{d-1}} W_0(t,x,\eta)u_0(x + \varrho(t,x,\eta))J(t,x,\eta)d\eta + \int_{\mathbb{S}^{d-1}} W_1(t,x,\eta)u_1(x + \varrho(t,x,\eta))J(t,x,\eta)d\eta.$$

The displacement appears in the SELFWARP component of our architecture while $\eta$ labels the heads. As the equation is linear, $\Gamma_{-t}$ does not depend on $u_{0,1}$.

Choosing (for a large $H$) the quadrature nodes $\eta_h \in \mathbb{S}^{d-1}$ and corresponding weights $w_k \in \mathbb{R}$, $h = 1, \ldots, K$, we can approximate integrals over $\mathbb{S}^{d-1}$ by a quadrature formula

$$\int_{\mathbb{S}^{d-1}} F(\eta)d\eta \approx \sum_{h=1}^{H} w_h F(\eta_h). \tag{24}$$

By using the quadrature (24) on $\mathbb{S}^{d-1}$ to approximate the above integrals, we obtain

$$\begin{aligned}
u(t,x) &\approx \int_{\mathbb{S}^{d-1}} W_0(t,x,\eta)u_0(x + \varrho(t,x,\eta))J(t,x,\eta)d\eta + \int_{\mathbb{S}^{d-1}} W_1(t,x,\eta)u_1(x + \varrho(t,x,\eta))J(t,x,\eta)d\eta \\
&\approx \sum_{h=1}^{H} W_0(t,x,\eta_h)u_0(x + \varrho_{t,h}(x))J(t,x,\eta_h)w_h + \sum_{h=1}^{H} W_1(t,x,\eta_h)u_1(x + \varrho_{t,h}(x))J(t,x,\eta_h)w_h,
\end{aligned}$$

where $\varrho_{t,h}(x) = \Gamma_{-t}(x,\eta_h) - x$ are the displacement fields corresponding to the head $\eta_h$.

# F. Related work on neural PDE learning

Early learning-based PDE solvers largely adopted CNN image-to-image backbones (Zhu & Zabaras, 2018; Bhatnagar et al., 2019; Pant et al., 2021), with U-Nets (Ronneberger et al., 2015) remaining a strong baseline due to their locality and multiscale structure. A parallel line of work frames PDE prediction as learning *solution operators* between (discretizations of) function spaces (Kovachki et al., 2023; Lu et al., 2021; Li et al., 2020). Within this operator-learning view, Fourier Neural Operators (FNOs) (Li et al., 2021; 2023a) have been particularly influential: they use spectral mixing to realize global interactions and have become a standard reference point across benchmarks.

A complementary family of operator-learning approaches uses an encoder-decoder factorization in which the solution map acts on a low-dimensional latent representation of the input field (Bhattacharya et al., 2021; Hesthaven & Ubbiali, 2018; Adcock et al., 2024; Franco et al., 2026; Serrano et al., 2023).

Following the success of Vision Transformers (Dosovitskiy et al., 2021), attention-based architectures have also been adapted to operator learning (Cao, 2021; Li et al., 2023b). Their set-based formulations facilitate irregular meshes and complex geometries (Hao et al., 2023; Wu et al., 2024; Luo et al., 2025; Wen et al., 2026), but attention typically trades away strong locality priors and incurs quadratic scaling in the number of tokens, which is restrictive at high spatial resolution and in 3D. Hybrid models that mix multiple formalisms also exist (Guibas et al., 2022), aiming to combine global coupling with practical efficiency.

More recently, several works pursue foundation-model training for scientific dynamics: pretraining on diverse PDE families followed by task-specific finetuning (Subramanian et al., 2023; Herde et al., 2024; McCabe et al., 2024; Hao et al., 2024). These efforts are enabled by standardized benchmark suites and data collections, including PDEBench (Takamoto et al., 2022), PDEGym (Herde et al., 2024), and The Well (Ohana et al., 2024). In contrast to convolution-, Fourier-, or attention-based mixing, FLOWERS build global interactions from sparse, state-dependent *warps* (pullbacks) coupled with pointwise channel mixing, providing an explicit transport primitive that complements these prior approaches.

