# OpenReview forum: "Flowers: A Warp Drive for Neural PDE Solvers"
_ICML.cc/2026/Conference — ICML 2026 spotlight_

### Official Review · Reviewer_SNg9 · 2026-02-16

**Soundness:** 4
**Presentation:** 4
**Significance:** 4
**Originality:** 4
**Overall Recommendation:** 5
**Confidence:** 4

**Summary:**

This paper introduces Flowers, a neural PDE solver architecture built entirely from multihead coordinate warps (pullbacks). Each head predicts a pointwise displacement field and samples input features at the displaced coordinates, inducing nonlocal interaction through sparse sampling at linear cost. The design is theoretically motivated from three complementary physics perspectives: flow maps for conservation laws, ray tracing for waves in inhomogeneous media, and a kinetic-theoretic continuum limit. Embedded in a U-Net scaffold, a compact 17M-parameter Flower achieves state-of-the-art performance across 16 diverse 2D and 3D PDE benchmarks from The Well and PDEBench, outperforming FNO, CNextU-Net, and attention-based baselines, while a 150M-parameter variant competes with much larger foundation models.

**Compliance With Llm Reviewing Policy:**

Affirmed.

**Final Justification:**

Rebuttal responses address all my concerns.

**Key Questions For Authors:**

1. Can you provide a theoretical or empirical analysis of how multiscale composition (U-Net scaffold) relates to PDE structure? The single-scale theory (characteristics, rays) is clean, but the actual architecture relies heavily on the multiscale scaffold (Table 8 shows it matters). A "multiscale characteristics" or "multiscale transport" interpretation would strengthen the paper.

2. What happens on strongly diffusion-dominated problems where no characteristic/ray structure exists? The paper notes Flowers underperform Poseidon/Walrus on Gray-Scott and active matter. Have you tried problems like pure heat equation or Stokes flow to probe the limits of the warp primitive?

3. Can you report results with confidence intervals (e.g., 3 seeds) on a representative subset of benchmarks? Even 5-6 datasets would suffice to assess variability.

4. The learned displacement fields (Figure 6) align with fluid velocity for shear flow. Does this alignment persist across physics regimes? For example, do displacements align with characteristics in compressible flow, or with wave propagation directions in acoustic scattering?

5. What is the sensitivity to the number of heads $H$? The ablation (Table 3) shows single-head hurts rollouts. Is there a saturation point? Does the optimal $H$ depend on the number of characteristic families in the PDE?

**Limitations:**

Yes. Section 5.1 is among the most honest limitations discussions I have reviewed. It explicitly identifies: (1) incomplete understanding of why warps work for non-hyperbolic physics; (2) that the theoretical vignettes are "intuition builders" not explanations; (3) that rollout stability is not solved; (4) that Flowers underperform foundation models on some diffusive problems. The paper does not overclaim. The only gap is that practical limitations (e.g., regular grid requirement, boundary condition handling) are not discussed, and the method appears tied to periodic/regular grids through the interpolation implementation, but this is not flagged.

**Strengths And Weaknesses:**

### Strengths:

- **S1: Physics-derived architecture primitive.** The multihead warp is not a generic ML construction; it is directly motivated by how solutions to conservation laws (characteristics) and wave equations (rays) propagate information. This is exactly the kind of science-to-architecture bridge that AI4Science papers should aim for.
- **S2: Exceptional empirical breadth.** State-of-the-art on 16 diverse benchmarks spanning acoustics, MHD, shallow water, astrophysics, turbulence, viscoelastic flows, and 3D dynamics. This is not a method that works on one narrow setting.
- **S3: Efficiency and scalability.** Linear cost in grid points (vs. quadratic for attention). Competitive or superior throughput to FNO and SCOT (Figure 7). Easily scales to 3D. A 17M model outperforms much larger baselines.
- **S4: Honest and insightful discussion.** Section 5.1 transparently discusses what is understood (why warps work for hyperbolic physics), what is not (non-hyperbolic physics, Helmholtz success), and what remains to be done (rollout stability). This level of intellectual honesty is rare and valuable.
- **S5: Three complementary theoretical perspectives.** Conservation laws (Section 3.1), geometric optics (Section 3.2), and kinetic theory (Section 3.3) each provide different intuitions for the same architecture. This makes the theoretical motivation robust and accessible to readers from different backgrounds.
- **S6: Interpretability.** Figure 6 shows that learned displacement fields align with physical velocity fields without explicit supervision. This provides evidence that the architecture discovers physically meaningful representations, not just data-fitting patterns.

### Weaknesses:

- **W1: Gap between theory and architecture.** The theoretical motivations (Section 3) concern single-scale, single-step operations (characteristics for short time, first-order flow map expansion). The actual architecture uses a U-Net multiscale scaffold with multiple stacked blocks. The paper does not rigorously connect the multiscale composition to any PDE structure. The kinetic continuum limit (Section 3.3) addresses depth but not the multiscale aspect.
- **W2: Incomplete understanding of non-hyperbolic success.** The paper is honest that Flowers' strong performance on diffusion-dominated problems (Gray-Scott, active matter) and the time-independent Helmholtz equation is not well explained by the theoretical motivation. This is flagged in Section 5.1 but not resolved. The suggestion that "warps can implement convolutions" (Appendix C) is interesting but ad hoc.
- **W3: Rollout stability remains a challenge.** While Flowers outperform baselines on 1:20 rollouts, several datasets show substantial error growth (e.g., active matter: 1.39 VRMSE at rollout vs. 0.025 one-step). The paper does not propose specific mechanisms for improving rollout stability, and this is the main practical limitation for deployment as a PDE surrogate.
- **W4: No error bars in main results.** Table 1 reports single best-validation-LR runs. While the standardized protocol is reasonable, the absence of confidence intervals across multiple seeds makes it harder to assess whether small differences (e.g., post_neutron_star_merger: 0.3269 vs. 0.3391) are meaningful.
- **W5: Missing comparison to mesh-based methods.** No comparison to MeshGraphNets, Transolver, or other geometry-aware methods that handle unstructured domains. While most benchmarks are on regular grids, the paper claims generality that would benefit from such comparisons.

---

> ### Author Rebuttal · Authors · 2026-03-31
>
> Thank you for the detailed, positive review, and in particular for the kind words about our limitations section.
> ## W1 and KQ1
> We agree that the significance of the U-Net scaffold (in particular in relation to PDE structure) should be better explained. Empirically, the single-scale Flower (Table 8 with new experiments: https://ibb.co/KzQMxbpS) already outperforms baselines on most datasets, confirming the warp primitive as the primary driver. The U-Net smooths patch-boundary artifacts and helps learn long-range correlations. Theoretically, the multiscale structure admits a natural multigrid V-cycle interpretation: restriction to coarser scales allows warps with larger relative displacements while prolongation transfers corrections to finer scales. Formalizing this (e.g., showing the composition corresponds to a multipole-type displacement decomposition) is an important follow-up. We will add a discussion in the revision.
> ## W2
> We believe the gap between hyperbolic and non-hyperbolic motivations is narrower than we initially wrote. Macroscopic diffusion arises as the hydrodynamic limit of kinetic transport in the frequent-collision regime. Our kinetic continuum limit (Appendix E, Eq. 18) is a Boltzmann-like equation, $\partial_t u + \eta \cdot \nabla_x u = Q(u)$. Deep Flowers can therefore represent diffusive dynamics. We emphasize that this is strictly (and rigorously) richer than the continuum limit of transformers, which yield a collisionless Vlasov equation (Geshkovski et al., 2025; Furuya et al., 2024). The collision term $Q$ in Flower limit is exactly what enables the kinetic-to-diffusive correspondence.
> One connection with Helmholtz may be via its Green's function which inherits the ray/geometric-optics structure of §3.2/Appendix F. The pullback representation carries over via Fourier transform in time, with "rays" becoming stationary-phase contributions to the resolvent. The full mathematical details are still work in progress but we will revise the manuscript with these arguments (§5.1).
>
> ## W3: Rollouts
> The quantitative picture is better than the informal phrasing in §5.1 suggests, which we will revise. Our updated Table 1 (https://ibb.co/5W6LRxp8) now includes rollouts for steps 21–60 and two additional PDEGym datasets, with relative VRMSE degradation for all four models (https://ibb.co/RGzcStYP). In 2D, Flower has second-lowest relative degradation and lowest median degradation in the 21–60 range. In 3D, it degrades less than CNextU-Net but more than FNO. Since Flower starts from significantly lower next-step error, comparable relative degradation still yields substantially better absolute rollout VRMSE. Models were trained on single-step prediction without pushforward or noise injection (per The Well's settings); the warp primitive is fully compatible with such strategies and this is indeed an exciting direction.
> ## W4 & KQ3
> We repeated experiments over two additional seeds (so three in total) on three representative datasets, for Flower and the respective second-best model on each (https://ibb.co/qzVrzKy). Next-step results are consistent across seeds; rollout scores show somewhat more variability.
> ## W5
> Please see **Limitations** below.
> ## KQ2
> Flower outperforms all baselines on diffusive problems (Gray–Scott, diffusion-reaction). No pure heat/Stokes datasets exist in common benchmarks, likely because these linear problems have very efficient spectral solvers, making neural solvers less appealing. Diffusion-reaction from PDEBench is the closest problem we tested, and Flower performs strongly. We now also offer a principled theoretical account (see **W2**).
> ## KQ4
> Additional visualizations: on shear flow, four heads co-align with the laminar flow direction (https://ibb.co/fYX7zM5q). On viscous fluid (https://ibb.co/s9V4YWWZ), one dominant head tracks primary flow while others capture secondary structure. On planetswe (1 head https://ibb.co/9HrVf3hf, 4 heads https://ibb.co/FqKR445j), fields are diverse, reflecting complex dynamics. Interpretability is clearest on laminar transport-dominated problems; on turbulent/multi-physics ones, displacements are structured but harder to interpret.
>
> ## KQ5
> We tested fixing total heads per layer and fixing channels per head (scaling to one head per channel), keeping parameters within 15–20M. On viscoelastic instability (https://ibb.co/fjvDJ0n), performance improves consistently with more heads, with no saturation. Memory reads scale linearly with heads; we recommend using as many as the throughput budget allows. Whether optimal *H* relates to the number of characteristic families is likely but not yet confirmed.
> ## Limitations
> We will make practical limitations explicit: periodic BCs wrap around; all others translate to zero for out-of-bounds queries. The method operates on structured grids, as do all baselines; no barrier exists for unstructured meshes, but this is beyond present scope.

---

> > ### Author Rebuttal · Reviewer_SNg9 · 2026-04-03
> >
> > Thanks for your rebuttal. This addresses all my concerns. I am keeping my score as-is.

---

> > > ### Author Response · Authors · 2026-04-07
> > >
> > > Thank you again for the expertly and encouraging review! Your questions really helped us improve the narrative. We will amplify the multiscale interpretation and the discussion of non-hyperbolic / kinetic-to-diffusive behavior. The suggested extra displacement analyses and multiple random seeds to get a sense of error bars indeed strengthen our arguments. (And again, we were very happy to read your kind words on our very explicit limitation section; we will add a note on meshes and boundary conditions.)

---

### Official Review · Reviewer_fJMN · 2026-03-01

**Soundness:** 4
**Presentation:** 3
**Significance:** 3
**Originality:** 4
**Overall Recommendation:** 5
**Confidence:** 4

**Summary:**

The paper introduces FLOWERS, a novel neural network architecture for learning PDE solution operators. It relies almost exclusively on learned, multi-head spatial warps called pullbacks instead of standard primitives like dot-product attention, convolutions, or Fourier layers, which can be expensive.
Operating entirely pointwise, the proposed SELFWARP layer uses a small local multi-layer perceptron to predict a set of coordinate displacements for each head. This models non-local spatial interactions by sparsely sampling the input feature map at these displaced locations. These lightweight warping mechanisms are integrated into residual blocks within a multiscale U-Net scaffold. Empirically, the authors demonstrate that FLOWERS achieve state-of-the-art forecasting accuracy and high computational throughput across a diverse suite of 2D and 3D operator-learning datasets such as The Well benchmark with their 150M-parameter model remaining highly competitive against much larger, so-called "foundation models".

**Compliance With Llm Reviewing Policy:**

Affirmed.

**Final Justification:**

The authors have addressed or committed to address my concerns. I believe this paper presents a compelling and interesting new method and viewpoint on established PDE problems and while it has many technical flaws (which hopefully the authors will address for the CRV), in my opinion this is a good addition to ICML and of interest to the SciML community.

I urge the authors to consider adding the multigrid viewpoint to the paper as I believe it connects this work to yet another large, classical body of methods and theory. I would especially like to see a more mathematical/rigorous treatment of the connection to AMG, SoC and similar heuristics. I believe this is interesting and will broaden the paper's audience to that community as well.

Therefore I recommend that the paper be accepted to ICML.

**Key Questions For Authors:**

Why did the authors use an overly-complex and custom notation system which is foreign to the ML community and requires much explanation, while simultaneously not utilizing this math to reach any rigorous conclusions? For example, the math could have been presented through the text of Appendix C and other materials in the appendices and then quickly cast as standard discretized linear algebra accessible to both ML and numerical methods practitioners? Please address this in the response. It is my belief that as it stands, the paper requires a substantial rewrite, the math substantially simplified, and language from the appendices moved into the main text. This reviewer is open to a clarification in the response, but as it stands this substantial need for a rewrite prompts me to err on the side of rejecting the paper while inviting the authors to rewrite it and resubmit it to another conference or the next ICML conference.

**Limitations:**

The authors list highly valid empirical limitations, such as degraded autoregressive rollouts and underperforming on highly diffusive problems compared to massive foundation models. However, the presentation lacks professionalism, relying on phrases like calling their model's performance "mystifying" and stating "we currently do not have a great explanation why", etc.
The limitations of the method are some that stem from the neural operator paradigm in general and this reviewer does not expect these discussed in detail in a short conference paper.

**Strengths And Weaknesses:**

### Soundness
**Strengths**: Empirically, the paper is robust. The authors evaluate their architecture across a diverse suite of 16 datasets from The Well, alongside additional benchmarks from PDEBench and WaveBench. The comparisons against FNO, CNextU-Net, and scOT are thorough and compelling. The authors show that smaller and computationally cheaper models can match or outperform larger models based on standard attention or Fourier mechanisms by constructing physics-informed models in the more literal sense of the word (not necessarily physics-informed in the PINN sense). Furthermore, the ablation studies effectively isolate the contribution of the multi-head warp mechanism, successfully proving that multiple heads are strictly necessary to stabilize autoregressive rollouts.

**Weaknesses**: The theoretical soundness in the main text is somewhat weak, with the authors openly admitting that their derivations are mere "intuition builders" rather than formal explanations. Furthermore, the model suffers from notable rollout degradation over long horizons (1:20 steps). The authors claim their physical motivation stems from hyperbolic transport and geometric optics, yet the main text fails to explain why the model performs exceptionally well on non-hyperbolic, diffusive problems like the Gray-Scott reaction-diffusion dataset and the time-independent Helmholtz equation. Some of this information is discussed more rigorously in appendices B, C, E and F, but the main text is written in an unrigorous manner and contains some unprofessional language for a scientific paper even by ML conference standards, which usually feature more relaxed and conversational writing styles.

### Presentation
**Strengths**: The visual aids are serviceable. Figure 1 provides a highly legible breakdown of the multi-head SELFWARP layer, and Figure 6 offers compelling qualitative evidence that the learned displacement fields physically align with underlying fluid velocities.

**Weaknesses**: The writing style is overly casual and highly unacademic for an ICML submission. The authors repeatedly rely on "weasel words" to wave away gaps in their understanding, describing their own model's performance as "mystifying" and explicitly stating "we do not understand". This informal tone masks the fact that the actual mathematical rigor, such as the Taylor expansion of the flow map, the derivation of the kinetic continuum limit, and the critical proof that pullbacks can approximate convolution integrals, is buried entirely in the appendices. The main text does not read as a rigorous scientific paper, and the authors must migrate these proofs to the main text to satisfy conference standards. The authors must rewrite much of the language in the paper to avoid ambiguous or subjective language such as "not good neural networks in an engineering sense" which is both too subjective and meaningless in the context of a scientific paper.

### Significance
The computational efficiency demonstrated by FLOWERS is compelling. By relying mainly on pointwise warps, the architecture maintains a linear computational cost with respect to grid points, allowing for seamless scaling to 3D problems, which  traditional neural PDE solvers often struggle with. Most significantly, the authors demonstrate that their 156M-parameter model (trained on 4 GPUs for a week) achieves performance competitive with a 1.3B-parameter foundation model (Walrus) that required 100 GPUs and two weeks of training. This parameter and compute efficiency makes FLOWERS a highly practical contribution to the SciML community.

### Originality
While the mechanical primitive itself, pointwise coordinate warping, is borrowed from existing computer vision techniques like spatial transformers and deformable convolutions, the originality lies in its isolation in the context of neural-operator-like PDE solvers. The authors strip away the dense, quadratic cost of dot-product attention and the rigid stencils of traditional convolutions, proving that learned pullbacks alone when embedded in a standard U-Net scaffold are a sufficient and highly effective spatial mixing primitive for PDE learning. It successfully replaces dense statistical routing with geometric, physics-aligned routing.
Furthermore, this approach can be elegantly conceptualized as a neural extension of Algebraic or Geometric Multigrid (AMG, GMG) methods. The multiscale U-Net structure acts as a traditional multigrid V-cycle, but rather than using static restriction and prolongation operators, the learned warps function analogously to AMG's "strength of connection" heuristics. By dynamically predicting displacement fields based on the local state, the network essentially builds physics-informed interaction stencils that route information along learned characteristics. This synthesis, a neural spatial multigrid driven by learned, characteristic-based routing, represents an original reframing of classical numerical methods and learned solvers and sits at an intersection of both that is a highly active field of research at the moment.

---

> ### Author Rebuttal · Authors · 2026-03-31
>
> Thank you for the constructive feedback and kind words on originality and significance. Below we address your concerns about presentation.
>
> # Writing style
>
> The reviewer raises two concerns: (1) informal language and (2) placement of mathematical content.
> On language: we agree that phrases like "mystifying," "not good neural networks in an engineering sense," are not common in mainstream academic prose. We propose concrete changes below. We do note that clearly stating what we do not understand is a deliberate choice motivated by transparency and does not signal incomplete work. With "we do not have a complete explanation" for Flower's success on a class of problems, we highlight the scope of our theory. While other reviewers value this candor (Reviewer SNg9: "among the most honest limitations discussions I have reviewed," "rare and valuable"), we appreciate that stylistic preferences vary and propose revisions that maintain intellectual honesty while using precise language.
> On rigor: as you comment, our work does not lack rigor. The Taylor expansion of the flow map (Appendix B), the kinetic continuum limit (Appendix E), and the proof that we recover convolution (Appendix C) are all detailed. The main text builds physical intuition connecting the architecture to characteristics, rays, and transport, with the formal development deferred to appendices for space. We agree, however, that this split may make the theoretical sections read as less rigorous than they are, and address this in the restructuring below.
> # Non-hyperbolic performance
> Please see the response to Reviewer SNg9, W2 for a discussion on theory that explains Flowers performance on diffusive problems. While the details are work in progress, we will add this discussion to the revised §5.1, replacing the informal language criticized by the reviewer. Independently, Gray–Scott and active matter are not purely diffusive; transport plays a role. (Also: Appendix C shows that flowers can represent non-adaptive integral operators.)
> # Notation and mathematics
> The reviewer suggests that the pullback formalism could be replaced by "standard discretized linear algebra." We believe this would obscure the connection with PDEs. The core operation, composing a feature field with a learned coordinate map, $u \mapsto u(x + \varrho(x))$ , is a continuous geometric operation: a nonlinear, state-dependent reindexing that acts on the argument of the function. In discretized form it leads to index combinatorics, $(u(x_j)) \mapsto (u(x_{\sigma(j)}))$, which are cumbersome to write in discretized linear algebra and would hide the connection to characteristics and flow maps. We also believe that readers of scientific ML tracks at ICML, and neural PDE work in particular, may appreciate the continuous notation as it foregrounds the relationship to PDE solution structure. Indeed, for a scalar conservation law (for example), the exact solution operator is a pullback along the characteristic flow (§3.1); wave equations have a similar structure (§3.2, Eq. 1).
> That said, we agree the main text should better connect the continuous formulation to the discrete implementation; see below.
> # Proposed revisions
> *Language cleanup.* We rephrase informal language: "mystifying" → "only partially understood"; "not good neural networks in an engineering sense" → "not neural networks with favorable inductive bias for learning PDE solution maps"; "rollouts which are currently not great" → "rollouts, where Flower achieves the strongest results but with clear room for improvement." We have done a thorough pass through the manuscript. (E.g.” rollouts which are currently not great", "wants to create vortices”, …, will all be rephrased.)
>
> *Restructuring §3.* Following Reviewer rqtx, we distill §3.3 to its key conclusion that stacking warps recovers a kinetic transport. We use the freed space to bring several “hard” results from appendices in the main text, add a "continuous to discrete" paragraph in §2 connecting pullbacks and grid-sampling, and properly introduce u(t,x), "characteristics," and "rays".
>
> *Replacing informal limitations.* We revise §5.1 with the kinetic-to-diffusive correspondence and Helmholtz–wave connection above.
>
> *Notation.* In §2.1, after the pullback definition, we add a remark showing the discrete equivalent: on a grid, (Id+ϱ)∗v reads v at displaced coordinates via bilinear interpolation, making the link to spatial transformers and deformable convolutions immediate.
>
> In case of acceptance, we dedicate the additional camera-ready page to this restructuring, ensuring the main text is self-contained on key theoretical results. We hope these revisions address your concern that too much rigor is in appendices, while keeping the physics connection.

---

> > ### Author Rebuttal · Reviewer_fJMN · 2026-03-31
> >
> > The proposed revisions seem acceptable. As noted, the mathematical and academic rigor exists in the paper but requires some amount of polish work to bring out. My concerns were only about the language used and I wish to state explicitly that discussing the limitations of the method and theory should never be to the detriment of a scientific paper.
> > That said, I urge the authors to consider the average ML reader who might not be able to understand the paper easily, and I hope the authors use the extra page well for this purpose and for the purpose of bringing more of the rigorous math forward.
> >
> > I thank the authors for their efforts in writing the original paper and the rebuttal and wish them good luck.

---

> > > ### Author Response · Authors · 2026-04-07
> > >
> > > We are happy to see that our rebuttal addressed your concerns. We in particular thought your multigrid / V-cycle interpretation is very interesting. We had related thoughts ourselves and we plan to mention that perspective in the revision. We will do our best to use the extra page well: make the paper easier to follow for a broader ML audience, strengthen the bridge between the continuous formulation and the implementation, bring more of the rigorous material into the main text, and smooth out the language throughout. Thank you again for your time, your constructive feedback, and the kind wishes.

---

### Official Review · Reviewer_XW1G · 2026-03-11

**Soundness:** 2
**Presentation:** 4
**Significance:** 2
**Originality:** 2
**Overall Recommendation:** 4
**Confidence:** 4

**Summary:**

Authors propose a novel architecture based on "warp layer". Roughly speaking, a warp layer is a pointwise linear layer (convolution with kernel size 1) combined with a composition operator. More specifically, for vector field $u(x)$, warp layer computes output $v(x)$ as follows:
1. Using $u(x)$ compute shift vector $s(x)$. Shallow MLP is used for this purpose.
2. Having a shift vector, find $u(x + s(x))$. If $u(x)$ is not available in $x + s(x)$ apply interpolation
3. Form final output as linear transformation $V u(x + s(x))$

The authors also introduce a multihead version of the warp layer (probe function in many distinct locations) and multiscale version of the architecture (use warp layer in place of convolution in the U-Net backbone).

Authors discussed three physics-based motivations for the warp layers: analogy with characteristics method for conservation laws, Green's function for wave propagation, and kinetic theory.

The resulting architecture is compared with several standard architectures on problems from The Well and PDEBench. The results indicate strong performance of proposed architecture in comparison to chosen baselines.

**Compliance With Llm Reviewing Policy:**

Affirmed.

**Final Justification:**

The authors supplied the additional ablation study I was interested in and provided extended explanations of the warp components, including improved visualization.

**Key Questions For Authors:**

**Novelty of warp operation, significance of architecture components.**

Multihead wrap layer first predict a set of neighbour locations based on the function value at a point and after that linearly transform the function evaluated at these points. The resulting operation is very close to standard convolution that does just that, with sampling in pre-defined locations. The other related architecture is Graph Kernel Network https://arxiv.org/abs/2003.03485 that uses Monte Carlo to evaluate integral $\\int k(x, y, u(x), u(y)) u(y) dy$, it is also available in the multiscale version https://arxiv.org/abs/2006.09535.

With that background in mind I kindly ask authors to address the following questions:

1. Why do authors believe the operation they introduce is fundamentally different from other already available local operations? For example, one may argue that the warp layer can be matched by a convolution layer. Can the authors discuss and refute this claim?

2. In my view the main feature of the proposed architecture is that positions, where functions are sampled, are adaptive and learnable. It is not clear whether this is significant or accurate operator learning or not:

    a. A valuable ablation study would be to generate locations randomly at the initialisation and train the resulting architecture with "frozen warp" layers. This experiment more directly demonstrates the importance of learning data-based displacement fields.

    b. Authors report results for warp architecture combined with U-Net backbone. Why is this necessary? Can a warp layer just learn to sample points further from target location automatically? It would be interesting to see how warp-based architectures perform without a U-Net backbone.

    c. Figure 6 seems to be the only place in the article where one can see the learned displacement field. Is it possible to report more results along these lines? I am particularly interested in seeing displacement fields for other heads, and some discussion of trained MLP that predict displacement locations (e.g., how sensitive this MLP is to the input, what is a typical range of prediction for displacement points, etc).

**Reported performance and comparison with other operators.**

In Table 1 authors report that their method performs better almost uniformly. From Appendix A one can learn that authors train all models themselves and this immediately raises suspicion. It is clear that there is little incentive for authors to investigate whether the baselines are tuned well enough.

I have several questions regarding this situation:

1. To make evaluation more objective I suggest to select a paper where independent evaluation of any baseline architecture was performed, train Flowers in the same setting and compare the performance with a well-tuned baseline. Preferably, the authors of that article should be invested in tuning all details of architecture to achieve best accuracy possible (e.g., they propose this particular architecture used as baseline).

2. In the section on Limitations authors report that "rollouts are currently not great". It does not seem this point is not mentioned anywhere else in the paper. The stability of rollouts is a crucial property, so I suggest expanding the discussion.

3. I also find it peculiar that the authors decide to limit training to 24 hours on H200 GPU. What is the reason for that? Does not this imply that at best the results they provide evaluate the speed of training, not the best possible accuracy of the network?

**Justification of warp layer.**

Authors provide several justifications for the construction of the warp layer and state that "we do not claim that any one of those perspectives causally explains the architecture’s empirical success." In my view the same examples can also be used in favour of convolution layers and graph kernel networks. Besides that, the disclaimer makes the reader question the significance of the explanation. Why provide this discussion at all, if it does not explain the performance and can be equally applied to other local architectures? I do not see why theoretical justifications are relevant and kindly ask authors to explain that.

**Misc.**

1. (page 1, lines 9-10, right column) "Most neural PDE backbones rely on Fourier mixing, convolutions, or attention." A large class of architecture that do not rely on such constructions are encoder-decoder architectures including: PCA-Net https://arxiv.org/abs/2005.03180, https://www.sciencedirect.com/science/article/abs/pii/S0021999118301190, https://arxiv.org/abs/2305.18642; neural field https://arxiv.org/abs/2306.07266, DOD https://arxiv.org/abs/2404.18841.

2. (page 1, lines 11-12, right column) "but through dense mixing that treats long-range interactions as equally plausible a priori" Can the authors explain the meaning of this claim and support it by theoretical or empirical results?

3. (page 1, lines 15-16, right column) "For many conservative phenomena $u(t,x)$ depends on local interactions between “channels” plus a ... " The sentence sounds slightly incoherent. One reason for that is that $u(t, x)$ suddenly appears without definition or explanation of what this field represents.

4. In Figure 1 authors use curvilinear mesh. What is the reason for that? It does not seem such a mesh is required for methods proposed by authors.

5. (page 3, lines 128-129, right column) Authors claim they use interpolation. How this interpolation is performed in the following cases: (i) complex geometry, (ii) prediction of displacement field falls outside of the domain, (iii) there is a shock present in the data, (iv) data is available on the unstructured grid? I think it is also interesting to evaluate the importance of interpolation (e.g., the order). For example, it is currently unclear how it affects the accuracy or rollout stability.

6. (page 3, lines 163-164, right column) "Our hope is that the reader will find at least one of the three vignettes helpful and illuminating, depending on." Sentence ends abruptly at the middle of the phrase.

7. In Section 4.2. authors mention they compare flowers with Walrus, but present no qualitative results. I kindly ask to supply a qualitative comparison of flowers and walrus accuracy in the main text or in the appendix.

**Limitations:**

yes

**Strengths And Weaknesses:**

**Soundness**

I find articles to be technically sound with several caveats. First, I have doubts that available empirical evaluation is trustworthy. Second, I do not find theoretical motivation presented by authors to be especially convincing. I will clarify these points in the next section of the review.

**Presentation**

In general, the article is easy to follow. Authors provide sufficient details and describe architecture well enough for interested readers to understand it and reproduce. In my view some terms are unnecessary (e.g., pullback notation, value map, warp), but they do not significantly compromise the clarity of the presentation.

**Significance and originality**

The primary contribution of the article is a novel architecture primitive for neural PDE solvers, called "warp layer". In my view, the warp layer is strongly related to the standard convolution operator and to the integral kernel, so I do not expect architectures based on the warp layer to be more expressive than standard convolution-based or "kernel-based" U-Net. These points are clarified below.

---

> ### Author Rebuttal · Authors · 2026-03-31
>
> # Q1: Significance/originality/novelty/justification of the warp layer
> You are right that pullback and convolutional networks are both universal. The same can however be said for MLPs (or even multivariate polynomials). The key is that for the operator classes we care about, pullbacks have a much more favorable inductive bias. PDE solution operators for nonlinear conservation laws naturally implement state-dependent coordinate changes (pullbacks along characteristics, rays, flow maps), simple to write as warps but difficult to approximate with fixed-kernel convolutions. That Flower consistently outperforms convolution- and attention-based models across 16+ benchmarks reflects this difference in inductive bias.
>
> More concretely: exactly as you point out, our layers differ from convolutions in adaptivity. An example: let ϱᵘ(x)=−2u(x), then $Fu=u(x−2u(x))$. For $u(x)=x$ this yields $Fu=−x$, a reflection. This simple pullback is very difficult to express as a fixed-kernel convolution $u→g∗u$. Our architecture directly implements the method of characteristics: for inviscid Burgers $u_t+uu_x=0$, the exact solution operator is a pullback along the characteristic flow, $u(x,T)=v(Φ^v(x))$ with $Φ^v(x)≈x-v(x)T$ for small $T$. This is a single warp, but expressing it as a convolution requires approximating a Dirac delta by a wide sum of indicator-weighted evaluations which is much less efficient. A more complicated example is incompressible 2D Euler, where vorticity at any time $t > 0$ can be written as a non-linear pullback of initial vorticity.
>
> Appendix C shows that pullbacks subsume convolution integral operators but the converse does not hold. Again, nonlinear, state-dependent transports are simple in terms of pullbacks but very complex as convolutional networks. ​​This is why the theoretical discussion is relevant: it identifies warps as a natural primitive for these dynamics. The disclaimer that these are "intuition builders" refers to the gap between the idealized single-step theory and the full multi-scale architecture, not to a lack of specificity to warps.
>
> Another lens on these differences from other primitives (e.g. attention) is our kinetic continuum limit (App. E). It shows that deep warp composition yields a Boltzmann-like transport equation whose hydrodynamic limit recovers diffusion, very different from the Vlasov limit of attention. Kindly see our response to Reviewer SNg9, W2 for details.
>
> # Q2: Suggested ablations
> (a) **Frozen warps.** Added to the ablation table (https://ibb.co/Sw3cCttd).
> (b) **Single-scale architecture.** See our response to Reviewer SNg9, KQ 1. In brief: single-scale Flower still outperforms tested baselines, confirming the primitive's effectiveness without the U-Net scaffold. The U-Net improves results further, mainly by smoothing artifacts at de-embedding, and facilitating the learning of long-range interactions.
> (c) **Displacement field visualizations.** See our response to Reviewer SNg9, KQ 4.
> # Performance/Baseline strengths
> Please refer to the discussion in our reply to Reviewer rqtx (especially the two PDEGym datasets + full Poseidon training) and to Reviewer SNg9 (on rollouts).
>
> **Training time.** The Well's benchmark prescribes 12h/H100 to “intentionally give an advantage to faster models” (The Well, App. E1). We relaxed it to 24h/H200/100 epochs which allows more models to converge while still rewarding efficiency. Completion rates across datasets (https://ibb.co/GQTzTNP0) show no link between timeout and Flower's advantage.
> # Misc
> 1. We will cite encoder-decoder architectures (PCA-Net, neural fields, DOD) in the revision; thank you for noting this omission.
> 2. By “dense mixing” we mean that Fourier layers correspond to global convolution kernels with unrestricted spatial support, and self-attention computes pairwise interactions across all positions with no built-in locality bias. We acknowledge this is more nuanced for structured positional encodings (e.g., RoPE).
> 3. We will rephrase to: “the evolution of the system state $u(t,x)$ depends on local interactions between conserved quantities plus upstream sources along characteristics or rays, in a state-dependent way.”
> 4. The curvilinear mesh in this figure is a visualization aid, not part of the method. We link a corrected figure (the original had a PDF export artifact): https://ibb.co/R4CFRJdL.
> 5. (i,ii, iv) See Reviewer SNg9, Limitations. (iii) Shocks are not treated specifically, but Flower performs well on datasets containing them (e.g., Euler). The multihead warps can plausibly implement characteristic selection (e.g. Hopf-Lax) required for shocks. Bilinear vs. bicubic interpolation showed no meaningful difference; we use bilinear for efficiency.
> 6. Corrected, thank you.
> 7. See our response to Reviewer rqtx, where we add a careful, new Poseidon comparison on periodic Euler under matched settings. Remarkably, even our 17M-parameter model outperforms (much bigger and much more trained) Poseidon on this task.

---

> > ### Author Rebuttal · Reviewer_XW1G · 2026-04-02
> >
> > I thank the authors for the clarifications and especially for the additional visualizations of the learned displacement field.
> >
> > I agree with the argument that the inductive bias of the architecture is important.
> >
> > In my view, the warp layer is a valuable addition to other operator learning approaches. My score is revised accordingly.

---

> > > ### Author Response · Authors · 2026-04-06
> > >
> > > First of all, thank you very much for the time and care you have invested in reviewing our paper and rebuttal. We sincerely appreciate it. Your questions about the distinction between pullbacks and convolutions / graph-kernel operators helped us sharpen the narrative + your questions about evaluation pushed us to clearly establish that the claims on broad state of the art performance are well supported. We are happy that the rebuttal resolved your concerns.
> > >
> > > We wanted to ask if there remains a specific issue that still underlies the current soundness, significance, and originality ratings, and the corresponding overall recommendation. As we understood your original review, the main concerns were (i) whether the warp layer is meaningfully distinct from existing local operators, (ii) whether the empirical comparisons were sufficiently strong and well supported, and (iii) whether learned displacements materially matter.
> > >
> > > In the rebuttal, we addressed these directly: for (i), with concrete examples and the formal inclusion discussion in Appendix C; for (ii), with matched-setting comparisons and additional supporting experiments; and for (iii), with the frozen-warp ablation and the displacement visualizations you requested. Your acknowledgement seems to indicate that these points were resolved. We were glad to read that you view the warp layer as a valuable addition to operator learning.
> > >
> > > We would be grateful to understand any remaining reservation that still supports the current component ratings and overall recommendation.

---

### Official Review · Reviewer_rqtx · 2026-03-14

**Soundness:** 3
**Presentation:** 2
**Significance:** 4
**Originality:** 3
**Overall Recommendation:** 5
**Confidence:** 4

**Summary:**

The paper introduces a novel architecture for solving PDEs. Instead of using architectures that capture global interactions via a Fourier-based operator, the author proposes using learned local deformations to model the PDE dynamics. Even though the architecture is simple, operations are mostly pointwise and use fewer parameters, it achieves better performance than others. The author also provides a theoretical intuition behind the effectiveness of their approach.

**Compliance With Llm Reviewing Policy:**

Affirmed.

**Final Justification:**

The questions are answered. And the authors promised to address the writing (presentation) issue in the final version.

**Key Questions For Authors:**

1. I see some of the datasets from The Well are skipped (e.g., post_neutron_star_merger). What is the reason?

2. Line 420-422, "indeed ... active matter" - I do not see the evidence that the proposed technique is not reaching the performance of Posidon. Please point to the table or the evidence. Furthermore, this raised the question of whether the proposed warping-based approach can be applied to large models such as Posidon. A hybrid architecture in which warping is applied within a foundation model has been explored in other fields [a]. It is crucial to discuss and test how such a hybrid approach performs in the domain of solving PDEs.

3. Line 144, the warping is modeled as $v(x + \varrho(x))$, which is ok for scaler valued function $v$. However, for a vector-valued function, the vector usually also undergoes a transformation. For example, in the case of the rotation of a vector-valued function, not only does the individual vector change position, they also goes through a rotation. Why, in the case of warping, is this not followed?

4. Can the authors think of any PDEs where the strictly local construction might fail?





a. Local Scale Equivariance with Latent Deep Equilibrium Canonicalizer

**Limitations:**

yes

**Strengths And Weaknesses:**

# Strength
The proposed approach is, to my knowledge, simple and well motivated. The results of this paper lead us to think that where global interaction is indeed necessary for solving benchmark PDE problems. Either local interaction is sufficient, or this indicates a severe limitation of the dataset. In either case, the work provides value to the research community. Furthermore, the technique is well motivated. The proposed model is simpler and faster, and performs better than the baselines.

# Weakness
**Experiments**
Although it is claimed to be efficient, Table 1 does not include the parameter counts for the different models.

As the proposed models are compact, a more appropriate application is to large-scale PDE problems (such as the car body problem dataset AhmedML or DrivAerNet++), where traditional models struggle to scale.

Also, no experiments were shown on complex geometry, such as an airfoil simulation, fluid-solid interaction, or Flowbench. It is unclear whether local deformation can accommodate complex geometries represented as a mesh.

Please see the question for more discussion on experiments.

**Paper Presentation**

a. L 15 -18 "For many ... way" : the notation $u$ terms like "characteristics" "rays" are suddenly introduced, hindering the readability

b. L 34-36 "Localized wave packets": No reference nor introduced earlier.

c. Even though sections 3.1 and 3.2 are understandable. I was unable to go through Section 3.3 without first reading Appendix E. The author might want to redirect the entire analysis to Appendix E, with only the final intuition clearly explained in the main text, accompanied by a figure.

---

> ### Author Rebuttal · Authors · 2026-03-31
>
> # On Results and comparisons with Walrus, POSEIDON, etc.
>
> [This addresses comments by several reviewers' on results, comparisons with other models and, strengths of baselines.]
>
> About yours and reviewer XW1G's concerns: we expanded Table 1 with Wave-Layer and CE-RM from the Poseidon paper. Beyond the scaled-down scOT matching The Well's 15–20M budget, we also train the original 40M-parameter scOT on both datasets. We keep the benchmark setup (24h/100 epochs, single H200, three learning-rate sweep), selecting the best configuration per model. Flower outperforms all baselines, including full-scale scOT, on both datasets (https://ibb.co/FcBZYv5).
>
> We expanded the main/benchmark results (https://ibb.co/5W6LRxp8) with the PDEGym datasets & longer rollouts (steps 21–60). For scaling/periodic Euler, we trained Poseidon under exact matched settings  (40 epochs,4→1, no time limit) https://ibb.co/mVXtBgc6. We find it remarkable that even Flower-Tiny (17M-parameter) outperforms Poseidon here, despite Poseidon's multi-domain pretraining advantage and it being the top model on this dataset per Walrus Table 1.
>
> PhysiX (arXiv:2506.17774) and PDE-FM (arXiv:2511.21861) also use The Well's 4→1 setting, making their own published numbers directly comparable with ours, though both are substantially larger and unconstrained by our wall-time/GPU budget limits. (NB: their reported baselines use The Well's original scores, now known to contain a bug; see this GitHub issue). Even so, Flower remains competitive. Against PDE-FM, our 17M Flower outperforms it on most shared datasets, often by large margins. PDE-FM wins on Rayleigh–Bénard, shear flow, TGC, and PNSM. Except for PNSM, these are large datasets where training for Flowers/baselines does not finish within our budget, suggesting the gap is compute- rather than architecture-driven. In general we do our utmost to ensure fairness.
>
> We also refer to Walrus (arXiv:2511.15684). Its Table 1 reports results for foundation models up to 1.3B parameters trained on ~100 GPUs for two weeks, with different input histories (6→1 in 2D, 3→1 in 3D). That Flower achieves comparable or superior accuracy to several of these configurations—trained from scratch with orders of magnitude less compute—suggests the warp primitive provides a strong inductive bias for PDE learning. A fully controlled comparison would require matching input protocols across all datasets (and possibly amounts of compute which are currently out of our reach).
>
> # Weaknesses
> 1. Parameter counts are currently in Appendix A.2. Models are scaled to 15-20M to match The Well's benchmark. We will add them to the table.
>
> 2. We appreciate this suggestion. We'd be eager to work with the mentioned datasets (AhmedML, DrivAerNet++), but our current implementation does not support general meshes. (There conceptually possible though; please see our response to Reviewer SNg9, limitations.)
>
> # Presentation
> a. In the revision, we will introduce u(t,x) as the state and explain "characteristics" and "rays" as curves along which information travels in hyperbolic physics; we will also revise the captions.
>
> b. We will add a clarification and a reference to Appendix F, where the geometric optics regime and wave propagation along rays are discussed in detail.
>
> c. This suggestion aligns with our planned restructuring of §3. We will distill it to its key intuition that stacking warp layers recovers a kinetic transport equation where heads index velocity directions, thus freeing space for material requested by several reviewers.
>
> # Key Questions
> 1. The reason for blank cells is that scOT does not support 3D; for all 3D datasets we report only FNO, CNUnet, and Flower. A clarifying comment has been added to Table 1.
>
> 2. +3 The claim uses Walrus, Table 1 (see discussion above). The SELFWARP primitive is a drop-in replacement for any spatial mixing layer and could be combined with attention or Fourier layers or within a larger model. This is promising but beyond our current scope which aims to establish warps as a standalone primitive. Rahman & Yeh (2025) use monotone coordinate maps for scale equivariance in vision transformers, which is related to differentiable spatial warping but differs substantially in both the mechanism and purpose from our approach.
>
> 3. You are right that for physical vector fields, a coordinate transformation requires both displacement and component transformation. In our architecture the warp acts in a lifted feature space where channels carry a transformed meaning; the appropriate transformation (suggested rotation of arrows) is then learned through pointwise mixing by the V matrix, rather than prescribed. This yields more flexibility.
>
> 4. Elliptic problems is where locality may be suboptimal since the solution depends on entire source distribution with no characteristic structure. That Flowers performs well on Helmholtz is interesting and a subject of ongoing research. (Cf. also hyperbolic--parabolic correspondence below.)

---

> > ### Author Rebuttal · Reviewer_rqtx · 2026-03-31
> >
> > The questions are answered. And the authors promised to address the writing (presentation) issue in the final version.

---

> > > ### Author Response · Authors · 2026-04-06
> > >
> > > First of all: thank you very much for the time and care you’ve invested in reviewing our paper and rebuttal. We sincerely appreciate it.
> > >
> > > We are happy to read that our response addressed all your concerns. In the light of this, we wanted to ask if there is a remaining issue that prevents you from updating your recommendation? From our reading of the discussion, your comments about parameter counts, matched comparisons, and presentation were addressed in the rebuttal. One thing we did not complete within the rebuttal window is a full mesh-based training study.
> > >
> > > We agree that arbitrary meshes and geometries are an important extension, but we view this as substantial follow-on work rather than a prerequisite for our present claims and results which are about a new kind of neural nets for regular-grid neural PDE learning. Within that scope the manuscript already presents a very thorough empirical evaluation across 16 2D and 3D benchmarks, including next-step and rollout results, a scaling study, and single-scale ablations; it also includes the acoustic-scattering-maze benchmark which shows a complex-geometry case. All this computational evidence clearly shows that Flowers perform substantially better than the strong baselines along multiple criteria.
> > >
> > > As a small illustration of the feasibility point we made in the rebuttal, we also verified in a simple prototype that the basic warp can be implemented on an irregular mesh (https://ibb.co/G4hkxCpX). We mention this as an illustration, to substantiate that there are no conceptual issues with meshes, not as an additional experiment. A strong trainable mesh model will require separate architectural work, tuning, and benchmarking.
> > >
> > > The reason we ask is that your acknowledgement suggests that the main issues were addressed, so we are unsure what remaining concern underlies the current recommendation. If there is something we have overlooked, we'd be grateful if you’d let us know.

---

### Decision · Program_Chairs · 2026-04-30

**Decision:**

Accept (spotlight)

**Comment:**

This paper proposes FLOWERS, a neural PDE solver architecture built around multi-head learned coordinate warps as the primary spatial mixing primitive. Reviewers agreed that this is a novel and meaningful architectural contribution, well aligned with the structure of transport- and wave-dominated PDEs. A major strength of the submission is its unusually broad empirical evaluation across diverse 2D and 3D benchmarks, where the method consistently achieves very strong accuracy and efficiency, often outperforming established convolution-, Fourier-, and attention-based baselines with substantially smaller models and lower compute.

The main concerns raised in the initial reviews were about presentation, the rigor and placement of the theoretical motivation, the distinction from related local operators, and the completeness of some empirical comparisons and ablations. The authors addressed these points well in the rebuttal by adding stronger matched-setting comparisons, additional ablations and visualizations, clarification of the role of the multiscale scaffold, and a more precise explanation of the inductive bias behind the warp primitive. Reviewers indicated that these responses resolved their major concerns.

Overall, the paper offers a technically strong, original, and practically relevant contribution to neural PDE solving. While the camera-ready version would benefit from polishing the exposition and bringing more of the key theoretical material into the main text, the submission is clearly worthy of acceptance.